# G4SPLAT: GEOMETRY-GUIDED GAUSSIAN SPLATTING WITH GENERATIVE PRIOR

**Junfeng Ni**[1,2,*]   **Yixin Chen**[2,†,✉]   **Zhifei Yang**[3]   **Yu Liu**[1,2]   **Ruijie Lu**[3]
**Song-Chun Zhu**[1,2,3]   **Siyuan Huang**[2,✉]

[*]Work done as an intern at BIGAI   [†] Project lead   [✉] Corresponding author
[1]Tsinghua University   [2]State Key Laboratory of General Artificial Intelligence, BIGAI   [3]Peking University

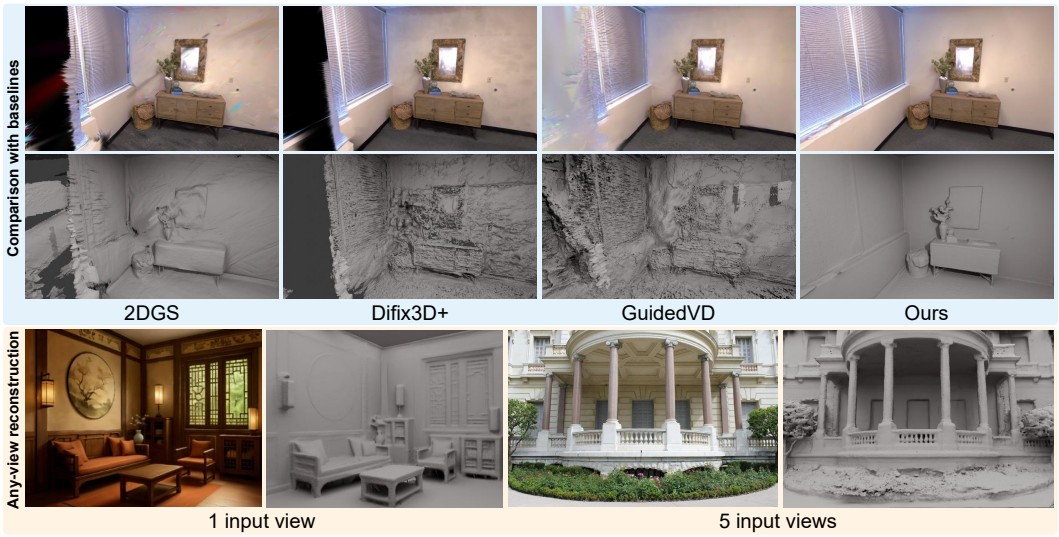

Figure 1: We propose **G4SPLAT**, which integrates accurate geometry guidance with generative prior to enhance 3D scene reconstruction. Our method significantly improves geometry and appearance reconstruction quality in both observed and unobserved regions (black areas in baselines), and generalizes well across diverse scenarios, including *single-view images* and *unposed videos*.

## ABSTRACT

Despite recent advances in leveraging generative prior from pre-trained diffusion models for 3D scene reconstruction, existing methods still face two critical limitations. First, due to the lack of reliable geometric supervision, they struggle to produce high-quality reconstructions even in observed regions, let alone in unobserved areas. Second, they lack effective mechanisms to mitigate multi-view inconsistencies in the generated images, leading to severe shape–appearance ambiguities and degraded scene geometry. In this paper, we identify accurate geometry as the fundamental prerequisite for effectively exploiting generative models to enhance 3D scene reconstruction. We first propose to leverage the prevalence of planar structures to derive accurate metric-scale depth maps, providing reliable supervision in both observed and unobserved regions. Furthermore, we incorporate this geometry guidance throughout the generative pipeline to improve visibility mask estimation, guide novel view selection, and enhance multi-view consistency when inpainting with video diffusion models, resulting in accurate and consistent scene completion. Extensive experiments on Replica, ScanNet++, DeepBlending and Mip-NeRF 360 show that our method consistently outperforms existing baselines in both geometry and appearance reconstruction, particularly for unobserved regions. Moreover, our method naturally supports single-view inputs and unposed videos, with strong generalizability in both indoor and outdoor scenarios with practical real-world applicability. Project page:
https://dali-jack.github.io/g4splat-web/.

# 1 INTRODUCTION

Recent advances in 3D Gaussain Splatting (3DGS) (Kerbl et al., 2023) have significantly improved photo-realistic novel view synthesis by enabling efficient 3D scenes reconstruction with dense view supervision. However, their performance degrades notably in sparsely captured settings due to insufficient geometric and photometric supervision. Depth regularization has been introduced to alleviate this issue (Turkulainen et al., 2025; Li et al., 2024a), but faithful reconstruction remains challenging because of the scale ambiguity inherent in monocular depth estimators (Yang et al., 2024). In addition, these methods are ineffective in under-constrained areas (*i.e.*, regions distant from or invisible to the input views), where no additional scene information can be provided.

More recent approaches (Liu et al., 2024d; Bao et al., 2025) leverage generative knowledge from pre-trained diffusion models (Blattmann et al., 2023; Yu et al., 2024b; Ma et al., 2025) to generate missing content in under-constrained regions. However, these approaches still struggle to faithfully reconstruct unobserved regions while maintaining consistency in the observed areas, as illustrated in Fig. 1. We identify two main factors underlying these problems. First, lacking reliable geometric supervision, these methods produce poor reconstruction quality even in observed regions with sparse input views, which undermines the geometric basis essential for inpainting unobserved areas. Second, these methods lack effective mechanisms to mitigate multi-view inconsistencies in diffusion model outputs, which lead to degraded scene recovery due to severe shape–appearance ambiguities (Zhang et al., 2020; Zhong et al., 2025).

In this paper, we argue that robust geometry guidance is fundamental to effectively employing the power of generative models and realizing high-quality reconstruction. To this end, we introduce **G4SPLAT**, which first leverages the prevalence of planar structures in man-made environments, consistent with the Manhattan world assumption (Coughlan & Yuille, 1999), to derive scale-accurate geometric constraints. Unlike matching-based methods (Wang et al., 2024a; Guédon et al., 2025) that often fail in non-overlapping regions, planar surfaces allow depth extrapolation: a 3D plane can be reliably estimated from partial depth observations and then extended across the entire surface. We leverage this property by aligning global 3D planes to obtain scale-accurate depth in planar regions and linearly adjusting monocular depth for unobserved non-planar regions, resulting in plane-aware depth maps that provide accurate geometric supervision in both observed and unobserved regions.

Beyond establishing improved geometry as the inpainting basis, we further incorporate geometry guidance into the generative refinement loop to alleviate the shape–appearance ambiguities. We begin by constructing a 3D visibility grid with scale-accurate depth, which provides more reliable visibility masks for inpainting compared to the noisy alpha maps. We also search for novel views by leveraging global 3D planes as object proxies, guiding the placement of novel viewpoints to maximize coverage of complete planar structures and thereby providing richer contextual cues for inpainting. Finally, we inpaint the novel views using a video diffusion model (Ma et al., 2025), while leveraging global 3D planes to modulate the color supervision for reducing cross-view conflicts.

Experiments on Replica (Straub et al., 2019), ScanNet++ (Yeshwanth et al., 2023), DeepBlending (Hedman et al., 2018) and Mip-NeRF 360 (Barron et al., 2022) demonstrate that our method consistently outperforms all baselines in both geometric and appearance reconstruction, with particularly strong improvements in unobserved regions. Ablation studies further confirm the effectiveness of each proposed component. Moreover, our approach supports reconstruction from casually captured videos or text-generated single images, as illustrated in Figs. 1 and 5, with significant potential in practical downstream applications in embodied AI (Huang et al., 2024b; 2025c; Jia et al., 2024), robotics (Li et al., 2024c;e; Zhi et al., 2025), and more (Jiang et al., 2024; Wang et al., 2024b).

Our main contributions are summarized as follows:

- We propose a novel method that leverages the plane representation to derive scale-accurate geometric constraints, substantially improving 3D scene reconstruction even in unobserved regions.

- We incorporate geometry guidance in the generative pipeline, which improves visibility mask estimation, novel view selection, and multi-view consistency with video diffusion models, yielding reliable and consistent scene completion.

- Extensive experiments demonstrate that our method achieves state-of-the-art performance on multiple datasets, with support for indoor, outdoor, single-view and unposed video reconstruction.

## 2 RELATED WORK

### 2.1 SPARSE-VIEW 3DGS

3D Gaussian Splatting (3DGS)(Kerbl et al., 2023; Lu et al., 2024a; Yu et al., 2024c) has demonstrated impressive reconstruction quality and training efficiency, but its performance deteriorates significantly when trained with only a few input views. To address this limitation, numerous approaches(Kumar & Vats, 2025; Zhang et al., 2024b; Wu et al., 2025b; Hong et al., 2025; Huang et al., 2025b; Xu et al., 2024; Paliwal et al., 2024; Zhao et al., 2024; 2025a; Bao et al., 2024; Liu et al., 2024f; Yin et al., 2024; Xiao et al., 2024; Lu et al., 2024b; Zheng et al., 2025) have been proposed. For example, DNGaussian (Li et al., 2024a) and FSGS (Zhu et al., 2024) incorporate depth regularization to suppress floaters in visible regions; yet, the scale ambiguity inherent in monocular depth estimators (Yang et al., 2024) prevents them from providing reliable geometric supervision. MAtCha (Guédon et al., 2025) attempts to overcome this limitation by introducing scale-accurate depth from structure-from-motion (SfM) methods (Wang et al., 2024a; Duisterhof et al., 2025). Nevertheless, it still struggles to reconstruct non-overlapping regions between input views. To address this issue, we propose to leverage the inherent extensibility of plane representations that propagate accurate depth estimates from overlapping regions to non-overlapping or even unobserved regions.

### 2.2 GENERATIVE PRIOR FOR 3DGS

Recent studies (Poole et al., 2022; Xiong et al., 2023; Weber et al., 2024; Wu et al., 2024; Liu et al., 2024e; Melas-Kyriazi et al., 2024; Shih et al., 2024; Paul et al., 2024; Cai et al., 2024; Chen et al., 2024b; Yu et al., 2024a; Ni et al., 2025) have demonstrated the effectiveness of leveraging diffusion models (Rombach et al., 2022; Liu et al., 2023) to provide powerful priors for 3D reconstruction. Some methods (Raj et al., 2025; Huang et al., 2025a; Chang et al., 2025) attempt to introduce 3D geometry priors into the reconstruction process. More recent approaches (Liu et al., 2024a;b;d; Zhao et al., 2025b; Gao et al., 2024; Zhou et al., 2025; Bao et al., 2025; Wu et al., 2025a;c; Fischer et al., 2025; Yin et al., 2025; Zhong et al., 2025) further employ video diffusion models (Blattmann et al., 2023; Yu et al., 2024b) to enhance cross-view consistency. However, since these models primarily rely on initial reconstructions to apply their generative power, they still suffer from numerous floating Gaussian artifacts and low-quality 3D geometry in their results, particularly in the inpainted regions, due to insufficient geometric constraints under sparse observations. To address this issue, we first apply scale-accurate depth supervision for both observed and non-observed regions to provide a robust geometric foundation, and integrate geometric guidance throughout the video diffusion models, leading to significantly improved reconstruction of both geometry and appearance.

### 2.3 PLANE ASSUMPTION IN RECONSTRUCTION

The Manhattan-world assumption (Coughlan & Yuille, 1999), and in particular the plane assumption, has been widely adopted in the reconstruction of man-made environments. Several methods (Liu et al., 2024c; Guo et al., 2024; Mazur et al., 2024; Liu et al., 2025b; Pataki et al., 2025) leverage this assumption in SfM and SLAM to improve matching accuracy, while other plane reconstruction approaches (Liu et al., 2019; Agarwala et al., 2022; Xie et al., 2022; Tan et al., 2023; Watson et al., 2024; Ye et al., 2025; Liu et al., 2025a) directly fit a set of planes to model indoor scenes. More recent studies (Guo et al., 2022; Li et al., 2024b; Chen et al., 2024a; Shi et al., 2025) incorporate the plane assumption to optimize 3D neural implicit representations (Mildenhall et al., 2020; Kerbl et al., 2023). For instance, GeoGaussian (Li et al., 2024d) and IndoorGS (Ruan et al., 2025) impose local planar constraints to regulate Gaussian splitting and movement, whereas PlanarSplatting (Tan et al., 2025) reconstructs 3D scenes as planar primitives directly from multi-view images. Different from these methods, our approach leverages the plane assumption to extract scale-accurate depth, which not only provides geometric guidance for optimizing the Gaussian representation but also facilitates the integration of generative prior, ultimately enabling precise scene reconstruction for both planar and non-planar structures across observed and unobserved regions.

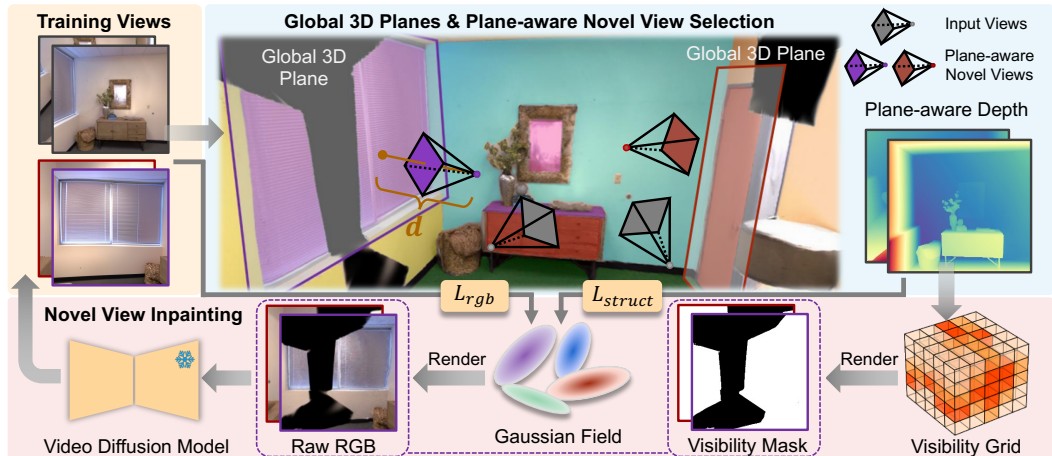

Figure 2: **Overview of G4SPLAT.** For each training loop (Section 3.4), we first extract global 3D planes from all training views and compute plane-aware depth maps (Section 3.2). Subsequently, we construct a visibility grid from these depth maps, select plane-aware novel views, inpaint their invisible regions, and incorporate the completed views back into the training set (Section 3.3).

## 3 METHOD

We propose **G4SPLAT**, a method that integrates accurate geometry guidance with generative priors to enhance 3D scene reconstruction. We begin by introducing the base model MAtCha (Guédon et al., 2025) and the overall training objective in Section 3.1. Next, we present our plane-aware geometry modeling in Section 3.2, followed by the geometry-guided generative pipeline in Section 3.3. Finally, we describe the overall training strategy in Section 3.4.

### 3.1 BACKGROUND

2D Gaussian Splatting (2DGS) (Huang et al., 2024a) extends the original 3D Gaussian Splatting (3DGS) framework (Kerbl et al., 2023) by collapsing 3D volumetric Gaussians into 2D anisotropic disks. Building on 2DGS, MAtCha (Guédon et al., 2025) introduces a chart alignment procedure that optimizes the chart parameters for each input view based on the outputs of MASt3R-SfM (Duisterhof et al., 2025). The chart parameters from all views are then used to generate scale-accurate depth maps, which serve as absolute depth supervision for training 2DGS.

Given N input images $\{I^i\}_{i=1}^N$ with its associated camera poses, the overall training objective of MAtCha combines an RGB reconstruction loss $\mathcal{L}_{\text{rgb}}$, the original 2DGS regularization loss $\mathcal{L}_{\text{reg}}$, and a structure loss $\mathcal{L}_{\text{struct}}$, formulated as:

$$\mathcal{L}_{\text{total}} = \mathcal{L}_{\text{rgb}} + \mathcal{L}_{\text{reg}} + \mathcal{L}_{\text{struct}}. \tag{1}$$

A more detailed description of each term is provided in the Appendix B.3.

However, the effectiveness of chart alignment heavily relies on accurate image correspondences; in regions with poor or missing matches, noticeable errors occur, as shown in Fig. 3a.

### 3.2 PLANE-AWARE GEOMETRY MODELING

**Per-view 2D Plane Extraction**  Inspired by prior work (Mazur et al., 2024; Ye et al., 2025), we assume that planar regions within an image exhibit consistent normal directions, smooth geometry, and similar semantics. Based on this assumption, we extract plane masks from images by combining instance masks from SAM (Kirillov et al., 2023) with normal maps generated from either depth-map gradients or a monocular predictor (Ye et al., 2024). Specifically, we apply K-means clustering to the normal map to obtain regions with coherent surface orientations. These regions are subsequently filtered using SAM masks, and only those assigned the same instance label and exceeding a predefined size threshold are treated as valid 2D plane masks. Example results are shown in Fig. 3a.

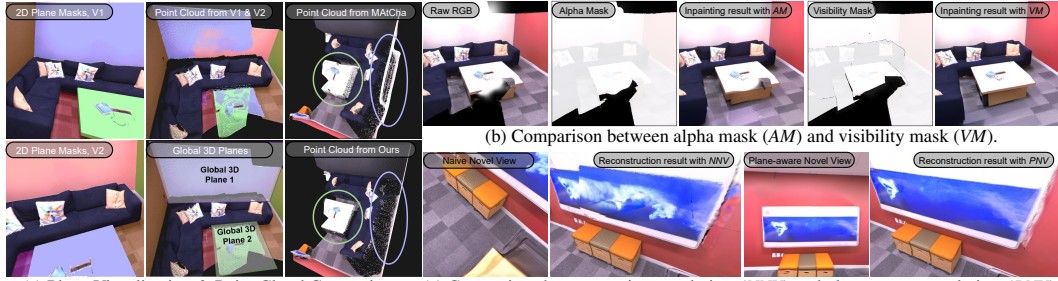

(b) Comparison between alpha mask (*AM*) and visibility mask (*VM*).

(a) Plane Visualization & Point Cloud Comparison.  (c) Comparison between naive novel view (*NNV*) and plane-aware novel view (*PNV*).

Figure 3: **Visualization of intermediate results.** Our method addresses key issues in prior approaches: (a) MAtCha produces noticeable errors in non-overlapping regions (highlighted by circles); (b) masks derived from alpha maps contain errors in visible areas of novel views; and (c) naive novel view selection offers only local coverage, causing visible seams in the final reconstruction.

**Global 3D Plane Estimation**    The 2D plane masks extracted from individual views are often oversegmented and lack global consistency, resulting in the same 3D plane being fragmented across multiple views. To address this issue, we leverage the 3D scene point cloud to establish correspondences among local masks and merge them into globally consistent 3D planes, as shown in Fig. 3a.

Specifically, for each per-view 2D plane mask, we collect the associated 3D points from the scene point cloud via projection. Two local planes are merged into the same global 3D plane if their associated 3D point sets exhibit sufficient spatial overlap and have similar normal directions. By repeating this process across all views, we obtain a set of global point collections $\{\mathcal{P}_k\}$, each representing a global 3D plane. Further details are in Appendix C.1. Each global 3D plane is represented as:

$$\Phi_k : \mathbf{n}_k^\top \mathbf{x} + d_k = 0, \tag{2}$$

where $\mathbf{n}_k \in \mathbb{R}^3$ is the unit normal vector and $d_k \in \mathbb{R}$ is the offset. To robustly estimate the plane parameters, we select a subset of high-confidence points $\mathcal{P}_k^{\mathrm{conf}} \subset \mathcal{P}_k$, defined as those observed in at least two different views. A 3D plane is then fitted to $\mathcal{P}_k^{\mathrm{conf}}$ by RANSAC (Fischler & Bolles, 1981):

$$\min_{\mathbf{n}_k, d_k} \sum_{\mathbf{p} \in \mathcal{P}_k^{\mathrm{conf}}} (\mathbf{n}_k^\top \mathbf{p} + d_k)^2, \quad \text{s.t. } \|\mathbf{n}_k\| = 1. \tag{3}$$

This procedure produces geometrically accurate and cross-view consistent 3D plane estimates, providing a reliable geometric basis for subsequent optimization.

**Plane-Aware Depth Map Extraction**    With the estimated global 3D planes, we extract a plane-aware depth map $D^v$ for each view $v$. Let $\{P_i^v\}_{i=1}^M$ denote the set of $M$ 2D plane masks in view $v$, each associated with a global 3D plane $\Phi_{k_i}$. For each pixel $\mathbf{u} \in P_i^v$, we cast a ray from the camera center $\mathbf{o}^v$ along the ray direction $\mathbf{r}^v(\mathbf{u})$, and compute its depth by intersecting the ray with the global 3D plane $\Phi_{k_i}$:

$$D_i^v(\mathbf{u}) = \frac{-\mathbf{n}_{k_i}^\top \mathbf{o}^v - d_{k_i}}{\mathbf{n}_{k_i}^\top \mathbf{r}^v(\mathbf{u})}. \tag{4}$$

For the non-planar regions of $I^v$ that are visible in the input view, we retain the depth values estimated by MAtCha (Guédon et al., 2025); for those that are not visible, we adopt a pre-trained monocular depth estimator (Yang et al., 2024) to predict a relative depth map $\hat{D}^v$. This relative map is aligned to an absolute scale using the already computed plane region depths $\{D_i^v\}$ via a linear transformation:

$$D^v(\mathbf{u}) = a_v \hat{D}^v(\mathbf{u}) + b_v, \tag{5}$$

where the scale $a_v$ and offset $b_v$ are estimated by least-squares fitting over pixels belonging to planar regions. The resulting depth map $D^v$ integrates geometry-consistent plane depths with refined monocular predictions in unobserved non-planar areas, yielding a complete and scale-accurate plane-aware depth representation for each view. As illustrated in Fig. 3a, this plane-aware depth significantly alleviates errors in non-overlapping regions when compared to MAtCha.

### 3.3 GEOMETRY-GUIDED GENERATIVE PIPELINE

**Geometry-Guided Visibility**    As shown in Fig. 3b, existing methods rely on inpainting masks derived from alpha maps, which often introduce errors within visible regions and thus degrade the inpainting results. To mitigate this issue, we employ scale-accurate plane-aware depth to model scene visibility using a visibility grid. Specifically, we first determine the 3D boundaries of the scene based on the depth maps from all training views. The scene is then discretized into a voxel grid $\mathcal{G}$. For each voxel, we determine its visibility by projecting its center to each training view and checking whether it falls within the valid depth range. A voxel is marked as visible (*i.e.*, visibility value = 1) if it lies within the observable depth range in at least one view. This visibility assessment is performed in parallel for all voxels, ensuring efficient construction of the visibility grid.

With the visibility grid, we render the visibility map for a novel view using the corresponding GS-rendered depth map. Specifically, pixel-wise visibility is evaluated by casting a ray from the camera center through each pixel and uniformly sampling $Q$ points along the ray up to the rendered depth. The visibility value $v_q$ of each sample point is determined via nearest neighbor interpolation over the visibility grid $\mathcal{G}$. The final per-pixel visibility $V^v(\mathbf{u})$ is computed as

$$V^v(\mathbf{u}) = \prod_{q=1}^{Q} v_q, \tag{6}$$

indicating that a pixel is considered visible only if all $Q$ sampled points along its viewing ray are marked as visible in the visibility grid $\mathcal{G}$.

**Plane-Aware Novel View Selection**    As shown in Fig. 3c, naive novel view selection strategies, such as the elliptical trajectory around the scene center (Wu et al., 2025c), tend to provide only limited local coverage. Inpainting results from these novel views often introduce noticeable artifacts in the final reconstruction due to multiview inconsistencies. To address this issue, we propose a plane-aware view selection strategy that ensures complete coverage of objects for the selected views by leveraging global 3D planes as object proxies. Global 3D planes generally provide sufficient structural and textural cues for object awareness to enable reliable inpainting. For each global 3D plane, we use its centroid as the look-at target and search for a camera center among the visible grid centers in the visibility grid. The selection process is guided by three objectives: maximizing coverage of plane points, minimizing distance to the plane, and encouraging alignment between the viewing direction and the plane normal. More details are described in Appendix C.2.

**Geometry-Guided Inpainting**    For each novel view $v$, we render the raw RGB map $\tilde{I}^v$ (Eq. A2) along with the visibility mask $V^v$ (Eq. 6). We then employ a pre-trained video diffusion model (Ma et al., 2025), which takes input images as reference $\{I^i\}$, and $\{\tilde{I}^v, V^v\}$ as input to jointly inpaint the occluded regions across all views, producing the completed images $\{\hat{I}^v\}$.

Despite this joint inference, the inpainted results $\{\hat{I}^v\}$ from existing generative models still exhibit multi-view inconsistencies, which can introduce artifacts during training (Zhong et al., 2025). To address this issue, we supervise each region by primarily relying on color information from a single view. For planar regions, we choose the view that provides the most complete observation of the corresponding plane, ensuring consistent inpainting. For non-planar regions, we use the first view in which the region becomes visible, which helps reduce cross-view inconsistencies. A more detailed discussion is provided in Appendix C.3.

### 3.4 OVERALL TRAINING STRATEGY

Our training pipeline consists of two stages: an initialization stage and a geometry-guided generative training loop. This design ensures that reconstruction begins with reliable geometry and progressively improves both coverage and multi-view consistency.

In the initialization stage, we first apply chart alignment in MAtCha to obtain an initial depth map for each input view. From these depth maps, we estimate global 3D planes and compute plane-aware depth maps, as described in Section 3.2. The Gaussian parameters are then initialized from the resulting point cloud and optimized using these plane-aware depth maps, producing a baseline model with accurate geometry in the regions observed by the input views.

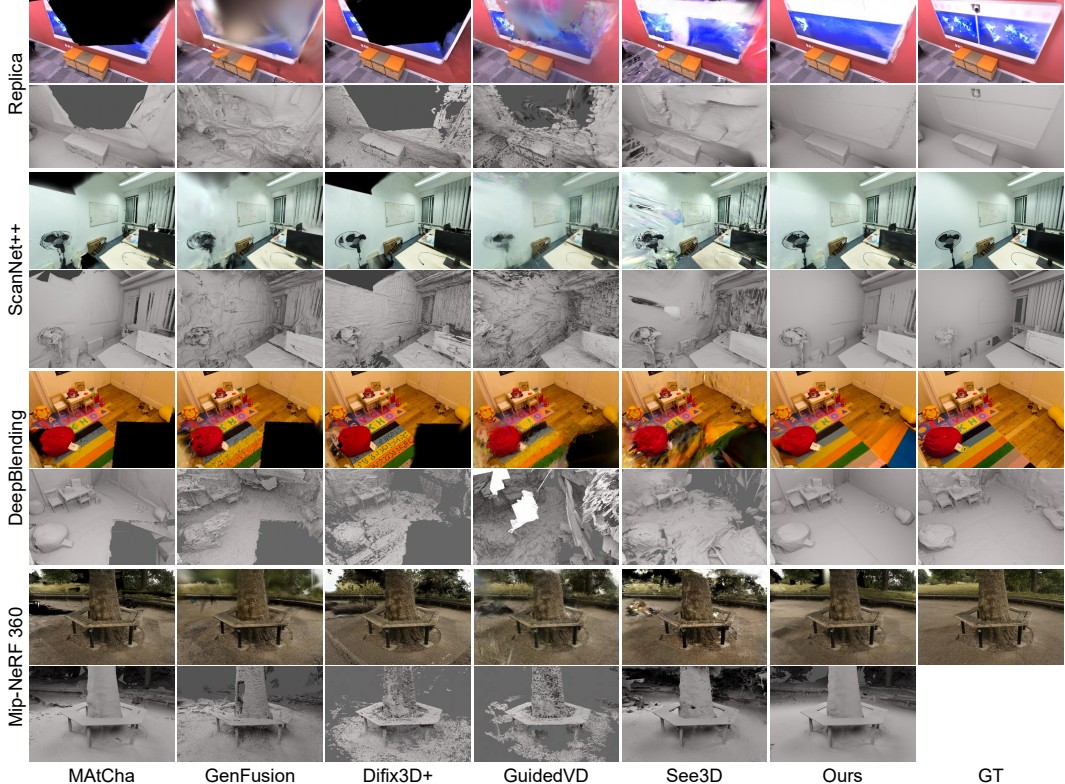

MAtCha        GenFusion        Difix3D+        GuidedVD        See3D        Ours        GT

Figure 4: **Qualitative comparison.** Our approach achieves better appearance and geometry reconstruction with fewer Gaussian floaters in both observed and unobserved regions.

The second stage refines and extends the reconstruction through the iterative process described in Section 3.3. As illustrated in Fig. 2, each loop begins by constructing a visibility grid from the current training views, followed by selecting novel viewpoints and inpainting their invisible regions; the inpainted novel views are then merged into the training set. Subsequently, global 3D planes and plane-aware depths are recomputed, and the Gaussians are fine-tuned with updated supervision. Repeating this loop progressively recovers unseen regions and corrects geometric misalignments.

During each round of 2DGS training, our method adopts the same total loss formulation as MAtCha (Eq. (1)), but enhances chart depth maps with plane-aware depth maps, thereby introducing stronger geometric constraints that lead to more accurate and consistent reconstructions. In our experiments, we employ three generative training loops; further details are provided in Appendix C.7.

## 4 EXPERIMENTS

We evaluate **G4SPLAT** on both geometric and appearance reconstruction in sparse-view 3D scenarios. Additionally, we present more experimental results in Appendix A, failure cases and discuss the method's limitations in Appendix D. For more extensive qualitative results and video demonstrations, please refer to our project page.

### 4.1 SETTINGS

**Datasets** We evaluate our method on both synthetic and real-world datasets. The synthetic dataset is Replica (Straub et al., 2019), which contains 8 indoor scenes. The real-world datasets include 6 scenes from ScanNet++ (Yeshwanth et al., 2023), 3 scenes from DeepBlending (Hedman et al., 2018) and 9 scenes from Mip-NeRF 360 (Barron et al., 2022).

Table 1: **Quantitative comparison from 5 input views.** Our method significantly outperforms all baselines across both reconstruction and rendering metrics. Top-3 results are highlighted as the first , second and third .

| Dataset | Method | Reconstruction | | | Rendering | | |
|---|---|---|---|---|---|---|---|
| | | CD↓ | F-Score ↑ | NC↑ | PSNR↑ | SSIM↑ | LPIPS↓ |
| Replica | 3DGS | 16.61 | 27.72 | 64.34 | 18.29 | 0.744 | 0.254 |
| | 2DGS | 14.64 | 48.01 | 74.14 | 18.43 | 0.735 | 0.306 |
| | FSGS | 18.17 | 26.87 | 64.16 | 19.19 | 0.766 | 0.259 |
| | InstantSplat | 21.00 | 19.67 | 62.01 | 19.39 | 0.762 | 0.255 |
| | MAtCha | 10.12 | 60.90 | 79.33 | 17.81 | 0.752 | 0.228 |
| | See3D | 12.74 | 45.27 | 73.98 | 19.22 | 0.735 | 0.328 |
| | GenFusion | 13.05 | 41.60 | 69.33 | 20.14 | 0.801 | 0.258 |
| | Difix3D+ | 13.71 | 43.11 | 65.34 | 19.42 | 0.779 | 0.231 |
| | GuidedVD | 27.87 | 17.29 | 61.64 | 22.51 | 0.822 | 0.260 |
| | Ours | 6.61 | 65.14 | 83.98 | 23.90 | 0.836 | 0.199 |
| ScanNet++ | 3DGS | 16.60 | 31.92 | 65.35 | 14.28 | 0.696 | 0.372 |
| | 2DGS | 14.34 | 51.97 | 70.01 | 13.91 | 0.661 | 0.429 |
| | FSGS | 23.80 | 27.86 | 64.53 | 14.80 | 0.731 | 0.362 |
| | InstantSplat | 21.32 | 25.44 | 60.67 | 15.02 | 0.742 | 0.355 |
| | MAtCha | 11.55 | 62.98 | 73.61 | 13.58 | 0.677 | 0.351 |
| | See3D | 13.03 | 53.65 | 70.39 | 14.76 | 0.684 | 0.426 |
| | GenFusion | 10.68 | 47.15 | 66.27 | 16.12 | 0.726 | 0.347 |
| | Difix3D+ | 13.15 | 53.91 | 67.30 | 14.09 | 0.701 | 0.340 |
| | GuidedVD | 25.35 | 16.67 | 60.48 | 17.90 | 0.807 | 0.336 |
| | Ours | 6.34 | 67.12 | 77.45 | 18.69 | 0.792 | 0.314 |
| DeepBlending | 3DGS | 31.44 | 20.02 | 55.39 | 15.33 | 0.571 | 0.489 |
| | 2DGS | 25.60 | 23.81 | 63.82 | 14.89 | 0.556 | 0.506 |
| | FSGS | 31.45 | 19.66 | 57.38 | 15.72 | 0.602 | 0.476 |
| | InstantSplat | 33.78 | 17.99 | 57.91 | 15.00 | 0.569 | 0.483 |
| | MAtCha | 22.36 | 26.80 | 67.92 | 14.74 | 0.558 | 0.465 |
| | See3D | 31.34 | 22.68 | 63.18 | 15.00 | 0.552 | 0.537 |
| | GenFusion | 30.70 | 22.37 | 58.70 | 16.20 | 0.626 | 0.468 |
| | Difix3D+ | 32.70 | 21.94 | 58.08 | 15.18 | 0.583 | 0.450 |
| | GuidedVD | 43.28 | 15.95 | 59.21 | 16.32 | 0.618 | 0.481 |
| | Ours | 20.72 | 28.02 | 72.04 | 16.76 | 0.645 | 0.440 |

For ScanNet++, DeepBlending, and Replica, we uniformly sample 100 images per scene. For the former two, we randomly select 5 images as input views and use the remaining 95 for testing. For Replica, we conduct experiments with 5, 10, and 15 input views to assess performance under varying view sparsity (Appendix A.3), while maintaining a consistent set of 85 test views across all settings. For Mip-NeRF 360, we follow the 9-input-view configuration as established in prior work (Wu et al., 2024; Gao et al., 2024).

**Metrics** Reconstruction quality is quantified using Chamfer Distance (CD), F-Score, and Normal Consistency (NC), while rendering performance is measured via PSNR, SSIM, and LPIPS. As the Mip-NeRF 360 dataset lacks ground-truth meshes, we evaluate only the rendering performance for those scenes; for the remaining three datasets, we report the full suite of metrics. Further details are provided in Appendices C.5 and C.6.

**Baselines** We compare our method with several representative baselines. These include the classical Gaussian Splatting approaches 3DGS (Kerbl et al., 2023) and 2DGS (Huang et al., 2024a), as well as state-of-the-art sparse-view 3DGS methods FSGS (Zhu et al., 2024), InstantSplat (Fan et al., 2024), and MAtCha (Guédon et al., 2025). We further evaluate against highly competitive recent approaches that integrate generative diffusion models into sparse-view 3DGS reconstruction, namely GenFusion (Wu et al., 2025c), Difix3D+ (Wu et al., 2025a) and GuidedVD (Zhong et al., 2025). In addition, we implement a variant of 2DGS augmented with the See3D (Ma et al., 2025). Notably, all baselines are augmented with MASt3R-SfM (Duisterhof et al., 2025) to provide robust geometric initialization and improve performance in sparse-view scenarios.

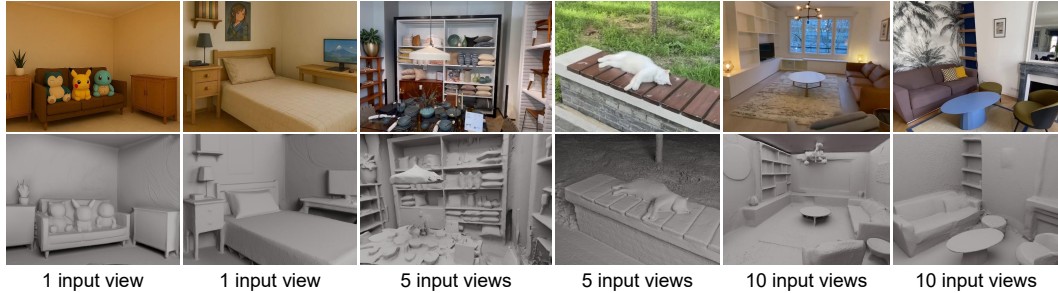

1 input view    1 input view    5 input views    5 input views    10 input views    10 input views

Figure 5: **Any-view scene reconstruction.** Our method demonstrates strong generalization across diverse scenarios, including indoor and outdoor scenes, unposed scenes and even single-view scenes.

## 4.2 RESULTS

**Novel View Synthesis Quality** By incorporating geometry guidance into the generative prior, our method achieves more accurate renderings in unobserved regions while producing fewer artifacts in observed regions, as illustrated in Figs. 4, A6 and A8 and quantified in Tables 1 and 2. In contrast, other methods that leverage generative prior exhibit notable limitations. For example, Difix3D+ attains relatively good quality in observed regions but fails to handle unobserved areas. GenFusion, See3D, and GuidedVD can hallucinate content in unobserved regions; however, their completions are blurry and compromised by severe floaters, and they even degrade the reconstruction quality of observed regions. These comparisons highlight the effectiveness of our geometry-guided generative pipeline in maintaining high fidelity across both observed and unobserved regions.

Table 2: **Quantitative comparison from 9 input views on Mip-NeRF 360.**

| Method | PSNR ↑ | SSIM ↑ | LPIPS ↓ |
|--------|--------|--------|---------|
| MAtCha | 16.44 | 0.453 | 0.389 |
| GenFusion | 17.82 | 0.452 | 0.439 |
| Difix3D+ | 16.28 | 0.389 | 0.426 |
| GuidedVD | 16.78 | 0.438 | 0.499 |
| See3D | 16.92 | 0.443 | 0.451 |
| Ours | 18.66 | 0.515 | 0.371 |

**Geometry Reconstruction Quality** As shown in Figs. 4, A6 and A8, Difix3D+, GenFusion, See3D and GuidedVD all suffer from severe shape–appearance ambiguities: even when the rendered views appear visually plausible, the reconstructed geometry is of poor quality. In comparison, our approach yields more accurate geometry in unobserved regions and produces smoother, floater-free reconstructions in observed regions. This improvement stems from our plane-aware geometry modeling, which provides reliable depth supervision for both observed and unobserved areas, ensuring consistency across the entire scene. Quantitatively, Table 1 shows that our method significantly outperforms all baselines on all reconstruction metrics across multiple datasets.

**Any-View Scene Reconstruction** As demonstrated in Figs. 1, 5, A1 and A2, our method exhibits robust performance across a wide range of scenarios, including indoor and outdoor scenes, single-view cases and unposed scenes such as YouTube videos. As shown in Table A1, our method consistently outperforms all baselines regardless of the number of input views. Moreover, in scenes with complex lighting, where existing baselines struggle to achieve accurate reconstruction even with dense input views, our approach leverages accurate geometry guidance to effectively suppress errors caused by significant brightness variations across viewpoints, thereby producing high-quality reconstructions, as shown in Fig. A2. In addition to achieving superior results on indoor scenes, as shown in the quantitative and qualitative results on the Mip-NeRF 360 dataset (Tables 2 and A3 and Figs. 4 and A7), our method also outperforms the baselines on outdoor, non-Manhattan, and less structured scenes. Furthermore, the *Museum* in Fig. 1 and the *Cat* in Fig. 5 further demonstrate its capability to handle arbitrary 3D structures.

## 4.3 ABLATION STUDIES

We conduct ablation experiments on Replica dataset to evaluate the contributions of the *generative prior* (GP), *plane-aware geometry modeling* (PM), and *geometry-guided generative pipeline* (PP). As shown in Table 3 and Fig. A4, we make the following key observations:

Table 3: **Ablation study.**

| GP | PM | PP | Reconstruction | | | Rendering | | |
|----|----|----|------|---------|------|-------|-------|--------|
|    |    |    | CD↓ | F-Score↑ | NC↑ | PSNR↑ | SSIM↑ | LPIPS↓ |
| × | × | × | 10.60 | 59.17 | 79.95 | 17.85 | 0.751 | 0.228 |
| ✓ | × | × | 9.46 | 56.99 | 77.58 | 19.63 | 0.740 | 0.295 |
| × | ✓ | × | 8.73 | 64.96 | 80.55 | 17.63 | 0.752 | 0.219 |
| ✓ | ✓ | × | 7.56 | 62.36 | 80.89 | 21.88 | 0.810 | 0.221 |
| ✓ | ✓ | ✓ | 6.61 | 65.14 | 83.98 | 23.90 | 0.836 | 0.199 |

Table 4: **Running time comparison.**

| Method | CD↓ | PSNR↑ | Time (min)↓ |
|--------|------|-------|-------------|
| MAtCha | 11.57 | 11.56 | 32.4 |
| See3D | 15.42 | 14.50 | 58.6 |
| GenFusion | 15.66 | 16.49 | 41.4 |
| Difix3D+ | 17.79 | 12.94 | 68.7 |
| GuidedVD | 24.29 | 19.02 | 141.4 |
| Ours | 8.77 | 20.26 | 73.3 |
| Ours (DS) | 9.33 | 19.36 | 43.5 |

1. Incorporating *generative prior* (*GP*) alone improves rendering quality but yields limited gains in geometry reconstruction. LPIPS, F-Score, and NC even decline, as *GP* alone tends to produce averaged, blurry results for the unseen areas. This indicates that directly introducing generative prior fails to perform as expected and leads to shape–appearance ambiguities. Moreover, as shown in Fig. A4, our method is compatible with different generative diffusion models, achieving strong performance across them.

2. Adding *plane-aware geometry modeling* (*PM*), either alone or in combination with *generative prior* (*GP*), significantly improves geometry reconstruction. Moreover, incorporating *PM* along with *GP* brings notable gains in rendering quality. This demonstrates that accurate geometry guidance effectively provides a clean geometry basis and suppresses Gaussian floaters, thereby enabling the generative model to fulfill its intended role.

3. Incorporating *geometry-guided generative pipeline (PP)* further enhances both rendering fidelity and geometric accuracy. This indicates that geometry guidance provides more accurate visibility masks, novel views with broader plane coverage, and consistent color supervision, thereby improving the generative process by mitigating multi-view inconsistencies.

### 4.4 DISCUSSION ON PLANE REPRESENTATION

Our method leverages a plane representation to obtain scale-accurate geometry guidance. We adopt this representation for three main reasons. First, planar structures are common in man-made environments and often occupy a large portion of the image, such as floors, walls, and tabletops. Second, the plane representation exhibits strong generalization. By fitting global 3D planes to accurate point clouds obtained in overlapping regions of input views, we can extend reliable depth estimation to non-overlapping or even unobserved planar regions. The accurate planar depth also enables linear alignment of monocular depth maps, improving depth accuracy of unobserved non-planar regions as well. Finally, the plane representation is simple, computationally efficient, and memory-friendly. As shown in Table 4, our method achieves high-quality reconstruction with a runtime comparable to other approaches that employ generative prior. Our accelerated variant, *Ours (DS)*, which downsamples the initial Gaussians, substantially reduces runtime while still outperforming all baselines.

Furthermore, our method maintains robust performance in non-planar or less structured scenes because it is a strict enhancement of the base model. This stems from our design that applies tailored supervision on regions according to their geometric characteristics: for *planar regions*, reconstruction is improved by leveraging accurate plane depth supervision across both observed and unobserved areas; for *non-planar regions*, observed regions preserve the base model's original supervision to ensure reliability, and unobserved portions benefit from linearly aligned monocular depth as an effective fallback where direct supervision is absent. Experiments on the Mip-NeRF 360 dataset validate these advantages. Additional discussions are provided in Appendices A.2 and C.7.

## 5 CONCLUSION

We present **G4SPLAT**, a geometry-guided generative framework for 3D scene reconstruction. The method first leverages plane representations to derive scale-accurate geometric constraints, and then integrates these constraints throughout the generative pipeline to effectively mitigate shape–appearance ambiguities. Extensive experiments demonstrate that our method consistently outperforms existing approaches in both geometry and appearance reconstruction, with notable improvements in unobserved regions. Moreover, our method naturally supports unposed video and single-view inputs, offering strong potential for practical applications in real-world scenarios.

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

APPENDIX

This appendix provides additional details and analyses that complement the main paper. Appendix A presents extended experimental results, including single-view and dense-view reconstruction experiments, performance under varying numbers of input views, and additional qualitative results. Appendix B offers further background information on our approach. Appendix C describes additional training and implementation details to facilitate reproducibility. Appendix D discusses typical failure cases and limitations of our method. Finally, Appendix E provides a statement on the usage of large language models (LLMs) during the development of this work.

## A  MORE EXPERIMENT RESULTS

### A.1  SINGLE-VIEW RECONSTRUCTION

For the single-view reconstruction experiment, we use an image generated by a text-to-image model (Esser et al., 2024) as input. We observe that inpainting-based video diffusion models, such as See3D (Ma et al., 2025), are less effective at synthesizing large unseen regions in a scene when only a single reference view is available. In contrast, camera-control video diffusion models like Stable Virtual Camera (Zhou et al., 2025) perform significantly better in this setting. A more detailed discussion on the choice of generative models is provided in Appendix C.4. Based on these observations, we adopt Stable Virtual Camera as the generative prior in our method: it is first employed to generate multiple novel views conditioned on the input single-view image, and these views are then used to train our model. Additional single-view reconstruction results are shown in Fig. A1.

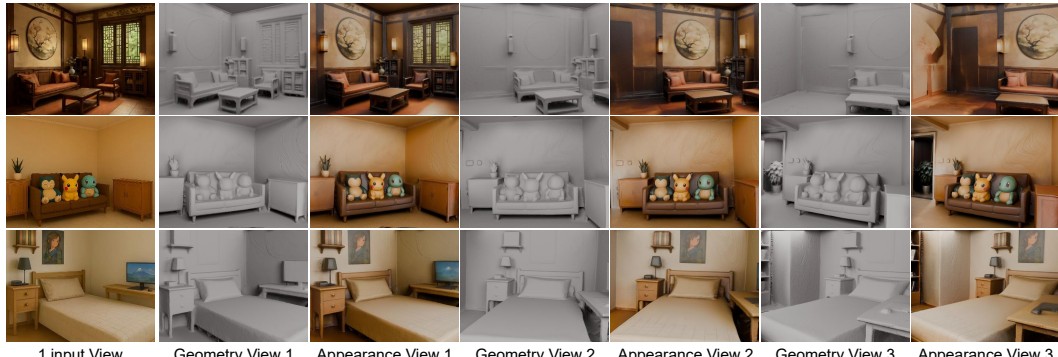

| 1 input View | Geometry View 1 | Appearance View 1 | Geometry View 2 | Appearance View 2 | Geometry View 3 | Appearance View 3 |

Figure A1: **More results of single-view reconstruction.** Our method achieves high-quality geometry reconstruction (*i.e.*, geometry view) and realistic texture recovery (*i.e.*, appearance view). The appearance view is obtained by rendering the exported colored mesh in Blender.

### A.2  DENSE-VIEW RECONSTRUCTION

Our method not only outperforms the baselines under the sparse-view setting, but also achieves superior results when dense input views are available. Methods based on Gaussian Splatting often struggle to reconstruct regions with strong specularities and reflections in scenes with complex lighting (Zhang et al., 2024a; Kulhanek et al., 2024). This difficulty arises because the color in such regions varies significantly across different viewpoints, making it challenging for explicit representations like Gaussian to fit them. In contrast, by incorporating scale-accurate geometry supervision, our approach effectively mitigates large color variations across views and produces accurate geometry reconstructions. As shown in Fig. A2, for a scene with 383 input views, our reconstruction results are substantially better than those of the baselines.

Although MAtCha can obtain reasonable depth supervision in chart views through chart alignment, it requires maintaining a separate chart representation for each chart view during the alignment process, as introduced in Appendix B.2. This results in high memory consumption, allowing only about 20 chart representations to be maintained on an NVIDIA 3090 GPU. In contrast, our method leverages global 3D planes, each stored compactly as a plane equation $(\mathbf{n}_k, d_k)$, which requires

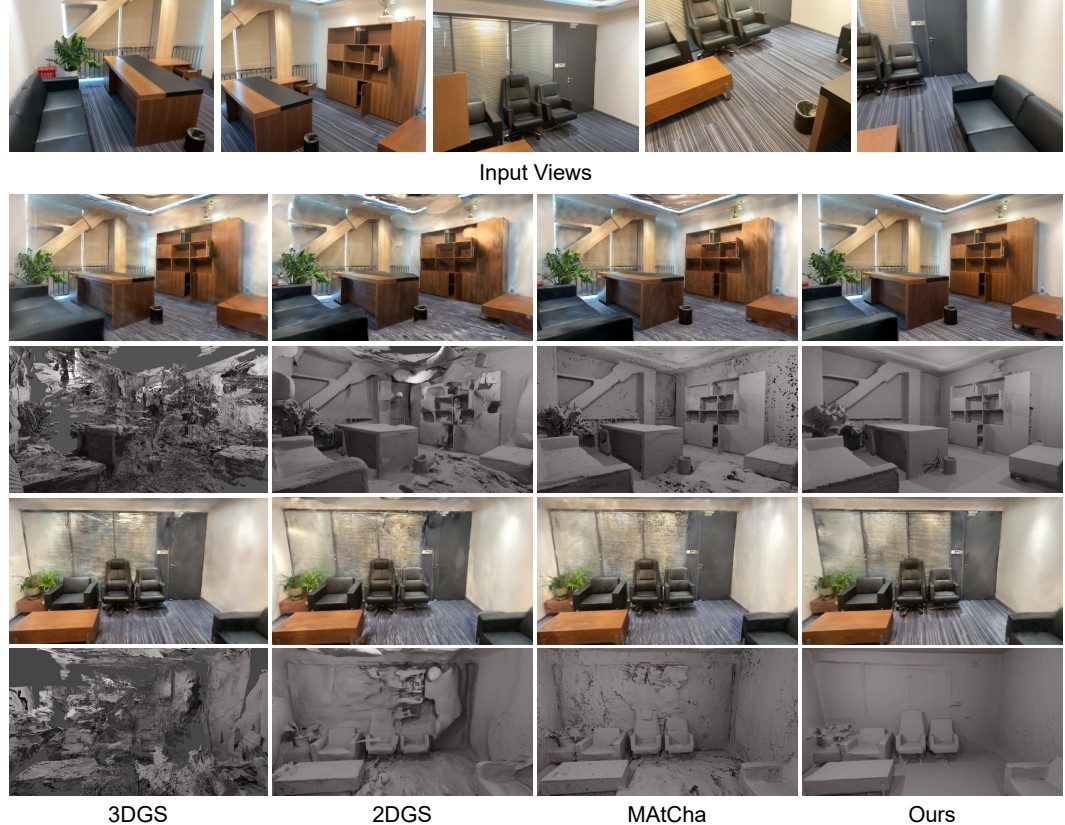

Figure A2: **Dense view reconstruction comparison.**

minimal memory overhead. Moreover, since all computations are based on projections, our approach can efficiently obtain scale-accurate depths for all input views by exploiting CUDA parallelization. All methods in Fig. A2 are evaluated on a single NVIDIA 3090 GPU.

### A.3 PERFORMANCE UNDER DIFFERENT NUMBERS OF VIEWS

We further assess the robustness of our method under varying numbers of input views. As summarized in Table A1, our approach consistently outperforms all baselines in both rendering quality and geometric accuracy, regardless of the number of input views. In particular, Fig. A2 illustrates that our method achieves markedly superior results when dense views are available, while Fig. A1 demonstrates that it still delivers strong reconstructions even with a single input view. These results highlight the effectiveness and generalizability of our design across diverse input-view scenarios.

Table A1: **Quantitative results on varying numbers of input views in the Replica dataset.** Our method consistently outperforms baselines regardless of the number of input views.

| Method | Reconstruction (CD↓ / NC↑) | | | Rendering (PSNR↑/LPIPS↓) | | |
| --- | --- | --- | --- | --- | --- | --- |
| | 5 views | 10 views | 15 views | 5 views | 10 views | 15 views |
| 2DGS | 14.64 / 74.14 | 9.37 / 81.24 | 7.17 / 85.33 | 18.43 / 0.306 | 21.79 / 0.207 | 25.30 / 0.139 |
| FSGS | 18.17 / 64.16 | 13.97 / 68.76 | 12.64 / 71.22 | 19.19 / 0.259 | 22.50 / 0.179 | 25.84 / 0.127 |
| MAtCha | 11.10 / 81.20 | 7.35 / 83.99 | 6.03 / 85.67 | 17.85 / 0.228 | 21.26 / 0.153 | 25.00 / 0.109 |
| See3D | 12.74 / 73.98 | 9.22 / 80.44 | 7.40 / 84.22 | 19.22 / 0.328 | 22.73 / 0.240 | 25.56 / 0.183 |
| GenFusion | 13.05 / 69.33 | 10.04 / 74.03 | 8.88 / 76.57 | 20.14 / 0.258 | 23.90 / 0.183 | 26.48 / 0.138 |
| Difix3D+ | 13.71 / 65.34 | 10.15 / 68.43 | 7.97 / 70.73 | 19.42 / 0.231 | 22.68 / 0.165 | 26.04 / 0.122 |
| GuidedVD | 27.87 / 61.64 | 20.30 / 64.53 | 16.62 / 68.54 | 22.51 / 0.260 | 25.63 / 0.205 | 27.91 / 0.163 |
| Ours | 6.61 / 83.98 | 4.88 / 85.49 | 3.98 / 87.25 | 23.90 / 0.199 | 27.48 / 0.140 | 30.22 / 0.094 |

### A.4 More Qualitative Results

Additional qualitative results are presented in Figs. A7 and A8. These results further demonstrate that combining geometry guidance with the generative prior effectively alleviates shape–appearance ambiguities, leading to significant improvements in both geometry reconstruction and rendering quality, particularly in regions that are unobserved by the input views.

## B Background

### B.1 Alpha Rendering in 2DGS

In 2DGS (Huang et al., 2024a), the value at a point $\mathbf{u} = (u, v)$ within the disk of a 2D Gaussian is defined by the Gaussian function:

$$g(\mathbf{u}) = \exp\left(-\frac{u^2 + v^2}{2}\right). \tag{A1}$$

Each 2D Gaussian is associated with an opacity $\alpha$ and a view-dependent color $\mathbf{c}$ represented using spherical harmonics. During rasterization, Gaussians are depth-sorted and composited into the final image using front-to-back alpha blending:

$$\mathbf{c}(\mathbf{x}) = \sum_{i=1}^{N} \mathbf{c}_i \alpha_i g_i\big(\mathbf{u}(\mathbf{x})\big) \prod_{j=1}^{i-1} \big[1 - \alpha_j g_j\big(\mathbf{u}(\mathbf{x})\big)\big], \tag{A2}$$

where $\mathbf{u}(\mathbf{x})$ denotes the $(u, v)$ coordinate obtained by intersecting the camera ray through pixel $\mathbf{x}$ with the corresponding 2D Gaussian disk.

Similarly, the depth map can be computed in the same manner, by replacing colors with the z-buffer values of the corresponding Gaussians.

$$d(\mathbf{x}) = \sum_{i=1}^{N} d_i \alpha_i g_i\big(\mathbf{u}(\mathbf{x})\big) \prod_{j=1}^{i-1} \big[1 - \alpha_j g_j\big(\mathbf{u}(\mathbf{x})\big)\big], \tag{A3}$$

where $d_i$ represents the z-buffer value of the $i$-th 2D Gaussian disk.

### B.2 Chart Representation and Alignment in MAtCha

MAtCha (Guédon et al., 2025) represents each chart with a lightweight deformation model designed to balance flexibility and efficiency. Each chart is parameterized by a sparse 2D grid of learnable features in UV space, together with a small MLP that maps these features to 3D deformation vectors. To properly handle depth discontinuities across object boundaries, the model is further augmented with depth-dependent features, allowing points at different depths to be deformed independently, even if they are close in UV space. This design preserves high-frequency details from the initial charts while capturing low-frequency, object-dependent deformations.

Each chart is initialized from a monocular depth map estimated by a monocular depth predictor (Yang et al., 2024). To ensure geometric consistency across views, MAtCha then performs an alignment stage that jointly optimizes chart deformations under multiple objectives.

**Fitting loss** To align charts with sparse SfM points, MAtCha minimizes their distances to the deformed charts:

$$\mathcal{L}_{\text{fit}} = \frac{1}{n} \sum_{i=0}^{n-1} \sum_{k=0}^{m_i-1} C_i(u_{ik}) \|\psi_i(u_{ik}) - p_{ik}\|_1 - \alpha \sum_{i=0}^{n-1} \log(C_i), \tag{A4}$$

where $\psi_i(u)$ denotes the deformed 3D position of UV coordinate $u$ on chart $i$, and $p_{ik}$ is the 3D position of the $k$-th SfM point visible in image $i$. The confidence map $C_i$ is learnable and downweights unreliable SfM points.

**Structure loss**  To preserve sharp geometric structures captured by the initial depth maps, MAtCha regularizes surface normals and mean curvature between the deformed charts and their initialization:

$$\mathcal{L}_{\text{struct}} = \sum_{i=0}^{n-1} \left( 1 - N_i \cdot N_i^{(0)} \right) + \frac{1}{4} \sum_{i=0}^{n-1} \| M_i - M_i^{(0)} \|_1, \tag{A5}$$

where $N_i$ and $M_i$ are the surface normals and mean curvatures of chart $i$, while $N_i^{(0)}$ and $M_i^{(0)}$ are their counterparts derived from the initial depth maps.

**Mutual alignment loss**  To enforce global coherence, MAtCha encourages neighboring charts to align by minimizing distances between projected overlapping points:

$$\mathcal{L}_{\text{align}} = \sum_{i,j=0}^{n-1} \sum_{u \in V_i} \min(\| \psi_i(u) - \psi_j \circ P_j \circ \psi_i(u) \|_1, \tau), \tag{A6}$$

where $V_i$ is the set of sampled UV coordinates on chart $i$, $P_j$ is the projection from 3D space to the UV domain of chart $j$, and $\tau$ is the attraction threshold that limits the maximum alignment distance.

**Final objective**  The overall optimization combines all three objectives:

$$\mathcal{L} = \mathcal{L}_{\text{fit}} + \lambda_{\text{struct}} \mathcal{L}_{\text{struct}} + \lambda_{\text{align}} \mathcal{L}_{\text{align}}, \tag{A7}$$

with $\lambda_{\text{struct}} = 4$ and $\lambda_{\text{align}} = 5$.

Although this alignment stage leverages cross-view projection constraints to yield more coherent scene geometry, its accuracy is ultimately limited by the quality of image correspondences. In regions with sparse or unreliable matches, it often introduces noticeable errors, motivating our plane-aware geometry modeling framework.

### B.3  Gaussian Surfels Refinement in MAtCha

To further enhance the reconstruction of fine-grained scene structures through differentiable rendering, MAtCha introduces a Gaussian surfel refinement stage based on 2DGS. In this stage, surfels are iteratively optimized using a joint loss function that combines photometric consistency with geometric regularization terms.

**RGB Loss**  The photometric loss is defined as a weighted combination of L1 loss and D-SSIM:

$$\mathcal{L}_{\text{rgb}} = (1 - \lambda)\mathcal{L}_1 + \lambda\mathcal{L}_{\text{D-SSIM}}, \tag{A8}$$

where $\lambda = 0.2$.

**Regularization Loss**  The regularization loss consists of a distortion loss and a depth-normal consistency loss, as introduced in 2DGS.

To prevent surfel drifting and enforce cross-chart consistency, the distortion loss is applied:

$$\mathcal{L}_d = \sum_{i,j} \omega_i \omega_j |z_i - z_j|, \tag{A9}$$

where $z_i$ denotes the intersection depth of the $i$-th surfel and $\omega_i$ is its blending weight.

To promote alignment of surface orientations across different charts, the depth-normal consistency loss is used:

$$\mathcal{L}_n = \sum_i \omega_i (1 - \mathbf{n}_i^{\mathrm{T}} \mathbf{N}_p), \tag{A10}$$

where $\mathbf{n}_i$ is the normal of the surfel and $\mathbf{N}_p$ is the normal derived from the depth gradient.

The overall regularization loss is then defined as:

$$\mathcal{L}_{\text{reg}} = \lambda_d \mathcal{L}_d + \lambda_n \mathcal{L}_n, \tag{A11}$$

with weight values $\lambda_d = 500$ and $\lambda_n = 0.25$.

**Structure loss** The structure loss follows a formulation similar to Eq. (A5):

$$\mathcal{L}_{\text{struct}} = \sum_{i=0}^{n-1} \|\bar{D}_i - D_i\|_1 + \sum_{i=0}^{n-1} \left(1 - \bar{N}_i \cdot N_i\right) + \frac{1}{4} \sum_{i=0}^{n-1} \|\bar{M}_i - M_i\|_1, \tag{A12}$$

where $\bar{D}_i$, $\bar{N}_i$, and $\bar{M}_i$ are rendered from the Gaussian surfels, while $D_i$, $N_i$, and $M_i$ are obtained from the charts.

**Total refinement loss** The complete optimization objective is given by:

$$\mathcal{L}_{\text{total}} = \mathcal{L}_{\text{rgb}} + \mathcal{L}_{\text{reg}} + \mathcal{L}_{\text{struct}}. \tag{A13}$$

During each round of 2DGS training, our method employs the same total loss formulation as MAtCha, but enhances it by computing $D_i$, $N_i$, and $M_i$ in the structure loss term using our plane-aware geometry modeling. This integration of more robust geometric constraints leads to more accurate and consistent reconstructions.

## C    MORE TRAINING DETAILS

### C.1    MERGING GLOBAL 3D PLANES

As described in Section 3.2, we merge 2D plane masks from different viewpoints by leveraging the 3D scene point cloud. To this end, we first compute the 3D point cloud corresponding to each 2D plane mask, and then evaluate the similarity between every pair of 3D point cloud sets using the overlap ratio. For computational efficiency, we project the entire 3D scene point cloud into all viewpoints at once. However, this projection process introduces two challenges: (1) along each camera ray, all points are projected onto the same pixel, which ignores occlusion; (2) when the surface points are insufficiently dense, some pixels may not receive projections from actual surface points but instead from points located behind the surface.

Both issues degrade the accuracy of global 3D plane merging. To address them, we first render the depth maps of all viewpoints using Gaussian surfels, and then back-project these depth maps to reconstruct the 3D scene point cloud. This ensures that every pixel in each viewpoint is associated with a valid surface point. Moreover, to account for occlusion, when projecting the reconstructed 3D scene point cloud into a given viewpoint, we compute the depth values of the points and require that their relative deviation from the Gaussian-rendered depth values does not exceed 1%.

These steps ensure that our global 3D plane merging process is both accurate and efficient.

### C.2    PLANE-AWARE NOVEL VIEW SELECTION

As described in Section 3.3, we formulate the selection of plane-aware novel view camera centers as a search problem. The objective is to maximize the coverage of plane points, minimize the distance between the camera and the plane, and encourage the camera viewing direction to align with the corresponding plane normal.

Let the camera center be denoted by $\mathbf{c}$, the look-at point by $\mathbf{p}$ (*i.e.*, the center of the 3D plane), and the plane normal by $\mathbf{n}$. The optimization problem is then defined as:

$$\mathbf{c}^* = \arg \max_{\mathbf{c} \in \mathcal{C}} \left( R(\mathbf{c}) + \left| \cos \theta(\mathbf{c}, \mathbf{p}, \mathbf{n}) \right| - D(\mathbf{c}, \mathbf{p}, \mathbf{n}) \right), \tag{A14}$$

where $\mathcal{C}$ denotes the set of visible voxel centers in the visibility grid, $R(\mathbf{c})$ denotes the ratio of plane points visible from $\mathbf{c}$ to the total number of plane points. $\theta(\mathbf{c}, \mathbf{p}, \mathbf{n})$ is the angle between the viewing direction $(\mathbf{p} - \mathbf{c})$ and the plane normal $\mathbf{n}$, and $D(\mathbf{c}, \mathbf{p}, \mathbf{n})$ denotes the distance from the camera center to the plane.

In addition to our proposed plane-aware novel view selection strategy, we incorporate an elliptical trajectory around the scene center, following prior work (Wu et al., 2025c), to further enhance view diversity.

### C.3 GEOMETRY-GUIDED INPAINTING

As shown in Fig. A3, training Gaussian representations with multi-view inconsistent inpainting results leads to black shadows in inconsistent regions, a phenomenon also observed in GuidedVD Zhong et al. (2025). To address this issue, GuidedVD constrains the diffusion denoising process to preserve observed regions, thereby alleviating inconsistencies near these regions. However, it remains ineffective for large missing areas (see Figs. 4 and A8) and suffers from slow training speed (see Table 4).

In contrast, our approach mitigates multi-view inconsistencies by incorporating scale-accurate geometry guidance throughout the generative pipeline (introduced in Section 3.3). Within this pipeline, we further introduce a strategy that modulates color supervision based on scale-accurate depth to reduce inconsistencies in inpainting results. Specifically, we first establish correspondences across views by depth projection before training the Gaussian representation. For each region, we primarily rely on color supervision from a single view: if the region lies on a 3D plane, we select the view that provides the most complete observation of that plane, ensuring consistent inpainting; for non-planar regions, we select the first view where the region is observed. These operations are executed in parallel via geometric projection, require only a single pre-processing step before Gaussian training, and thus incur minimal computational overhead.

As shown in Fig. A3, our approach substantially reduces the impact of multi-view inconsistencies, producing sharper and cleaner renderings.

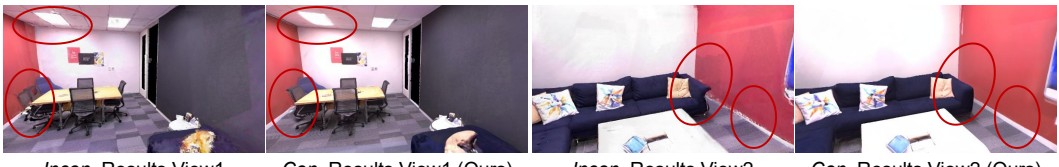

*Incon.* Results View1      *Con.* Results View1 (Ours)      *Incon.* Results View2      *Con.* Results View2 (Ours)

Figure A3: **Training results with inconsistent (*Incon.*) vs. consistent (*Con.*) multi-view inpainting.** Training Gaussian representations with inconsistent inpainting leads to black shadows in non-consistent regions, while our method significantly reduces such artifacts, yielding sharper and cleaner renderings.

### C.4 CHOICE OF GENERATIVE MODELS

For the generative models in our experiments, we primarily employ the inpainting-based diffusion model, See3D (Ma et al., 2025). Notably, our framework remains highly effective when integrated with alternative generative priors, such as ViewCrafter (Yu et al., 2024b), as evidenced by the quantitative and qualitative results in Table A2 and Fig. A4. Beyond these, our model is also compatible with camera-controlled diffusion models like ZeroNVS (Sargent et al., 2024) or Stable Virtual Camera (Zhou et al., 2025). In these configurations, the input images serve as references, while our selected novel views act as conditions. The visible masks can thus keep the visible areas unchanged while leveraging color supervision from viewpoints that observe the global 3D plane most completely. This capability is demonstrated in the single-view testing, as detailed in Appendix A.1.

Table A2: Quantitative results of our method integrated with different diffusion models.

| Method | Reconstruction | | | Rendering | | |
|---|---|---|---|---|---|---|
| | CD↓ | F-Score ↑ | NC↑ | PSNR↑ | SSIM↑ | LPIPS↓ |
| MAtCha | 10.12 | 60.90 | 79.33 | 17.81 | 0.752 | 0.228 |
| GenFusion | 13.05 | 41.60 | 69.33 | 20.14 | 0.801 | 0.258 |
| Difix3D+ | 13.71 | 43.11 | 65.34 | 19.42 | 0.779 | 0.231 |
| GuidedVD | 27.87 | 17.29 | 61.64 | 22.51 | 0.822 | 0.260 |
| Ours (w. ViewCrafter) | 7.95 | 64.19 | 81.82 | 22.08 | 0.812 | 0.210 |
| Ours (w. See3D) | **6.61** | **65.14** | **83.98** | **23.90** | **0.836** | **0.199** |

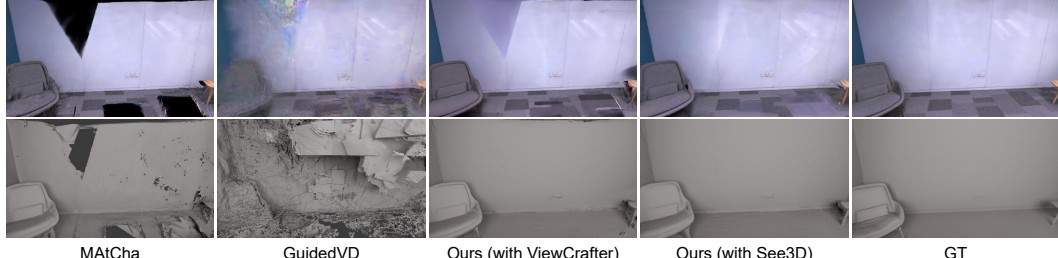

| MAtCha | GuidedVD | Ours (with ViewCrafter) | Ours (with See3D) | GT |

Figure A4: **Qualitative comparison from 5 input views using different diffusion models.** G4Splat produces high-quality, smooth geometry reconstruction even when paired with a weaker generative prior (*i.e. Ours with ViewCrafter*), and it continues to effectively suppress Gaussian floaters and reduce blurriness in the rendered results.

## C.5 DATASET DETAILS

For each scene, we uniformly sample 100 images. For ScanNet++ (Yeshwanth et al., 2023) and DeepBlending (Hedman et al., 2018), we randomly select 5 images as input views and use the remaining 95 images as test views. For Replica (Straub et al., 2019), we conduct three groups of experiments with 5, 10, and 15 input views to assess the performance under varying levels of view sparsity in Table A1. In all three Replica experiments, we use the same 85 test views to ensure a consistent evaluation set across settings.

## C.6 EVALUATION METRICS

**Rendering Metrics**   Following prior work (Guédon et al., 2025), we evaluate rendering quality using PSNR, SSIM and LPIPS computed with the VGG network (Simonyan & Zisserman, 2014).

**Reconstruction Metrics**   Following prior work (Yu et al., 2022; Guédon et al., 2025), we evaluate the Chamfer Distance (CD) in $cm$, F-score with a threshold of $5cm$ and Normal Consistency (NC) for 3D scene reconstruction. These metrics are defined in Table A4.

## C.7 IMPLEMENTATION DETAILS

We implement our model in PyTorch (Paszke et al., 2019) and conduct all experiments on a single NVIDIA A100 GPU, except for the dense-view reconstruction results in Appendix A.2, which are obtained on a single NVIDIA 3090 GPU. As described in Section 3.4, after completing the initialization stage, we perform three geometry-guided generative training loops. In each loop, 10 additional views generated by See3D are incorporated into the training set. Both the initialization stage and each loop train the Gaussians for 7000 iterations.

As shown in Table 4, training a single scene from Replica *room_0* with 5 input views using our method takes slightly over one hour. In addition, we evaluate a variant with a downsampled number of Gaussians, which further accelerates training while maintaining competitive performance, as reported in the *Ours (DS)* entry of Table 4.

## D  FAILURE CASES AND LIMITATIONS

In this section, we present and analyze representative failure cases. As shown in Fig. A5(a), our method is partially limited by the capabilities of current video diffusion models. For example, video diffusion models such as See3D (Ma et al., 2025) often struggle to ensure that the colors in the completed regions exactly match those of the original scene. This can lead to inconsistencies when training Gaussians with the completed images, resulting in rendering outputs that do not fully align with the visible surrounding areas. Nevertheless, by introducing scale-accurate geometry supervision, our method can achieve precise geometry reconstruction even from inconsistent completions, as demonstrated in Fig. A5(a).

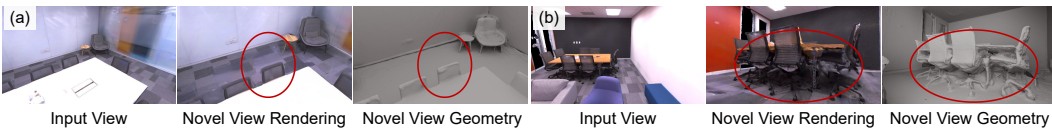

| Input View | Novel View Rendering | Novel View Geometry | Input View | Novel View Rendering | Novel View Geometry |

Figure A5: **Failure cases.**

Fig. A5(b) illustrates another limitation: our approach struggles with heavily occluded regions, such as the chairs partially blocked by a table. Because the table and the occluded regions of the chair are in close proximity, generating a reasonable novel camera view of the occluded areas is challenging. We hypothesize that incorporating object-level prior (Ni et al., 2025; Yang et al., 2025) could help reconstruct such severely occluded regions.

Moreover, our method relies on the plane representation to obtain scale-accurate depth. While the monocular depth estimator used for non-plane regions yields satisfactory results in our experiments, we believe that adopting a more general surface representation, which can naturally model both plane and non-plane regions, could yield more accurate depth, particularly in non-plane areas. Although such a representation may be less computationally efficient than plane-based approaches, it is expected to improve overall reconstruction quality.

## E   LLM USAGE STATEMENT

The authors acknowledge the use of Large Language Models during the preparation of this manuscript. LLMs were employed solely to improve clarity of expression and to refine grammar throughout the text. All AI-generated suggestions were carefully examined, refined, and validated by the authors. The research contributions, experimental design, data analysis, and scientific conclusions remain entirely the original work of the authors.

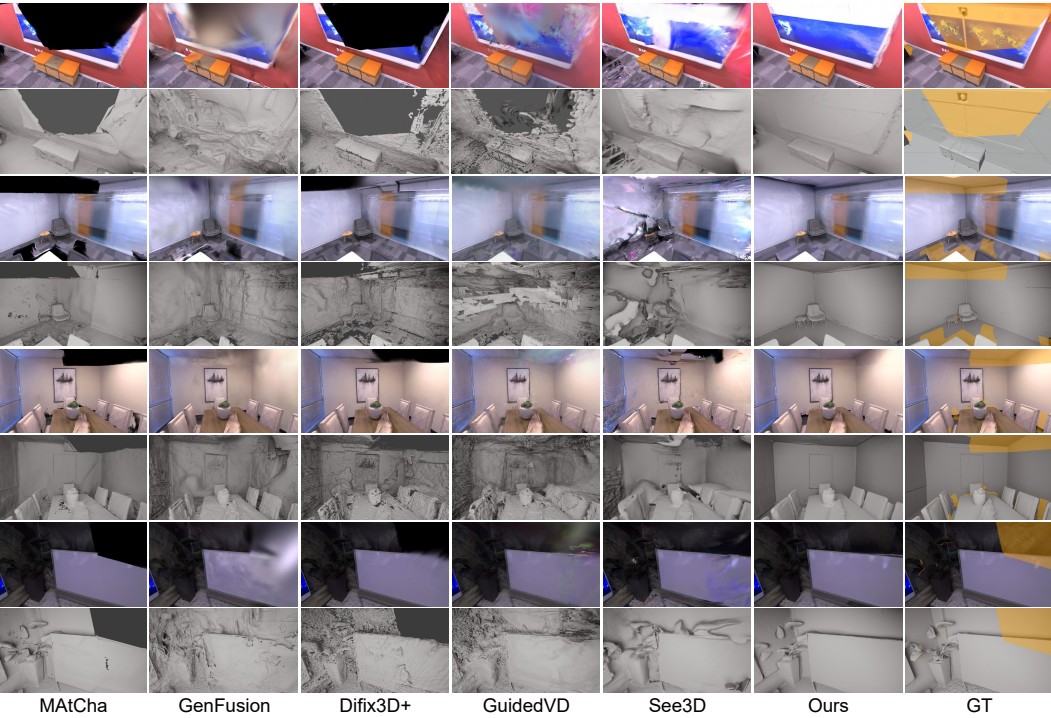

| MAtCha | GenFusion | Difix3D+ | GuidedVD | See3D | Ours | GT |

Figure A6: **Qualitative comparison from 5 input views with visibility mask.** In the GT images, golden regions indicate areas unobserved across the 5 input views. G4Splat outperforms baselines in both visible and unobserved regions, producing superior geometry with improved smoothness and minimal Gaussian artifacts.

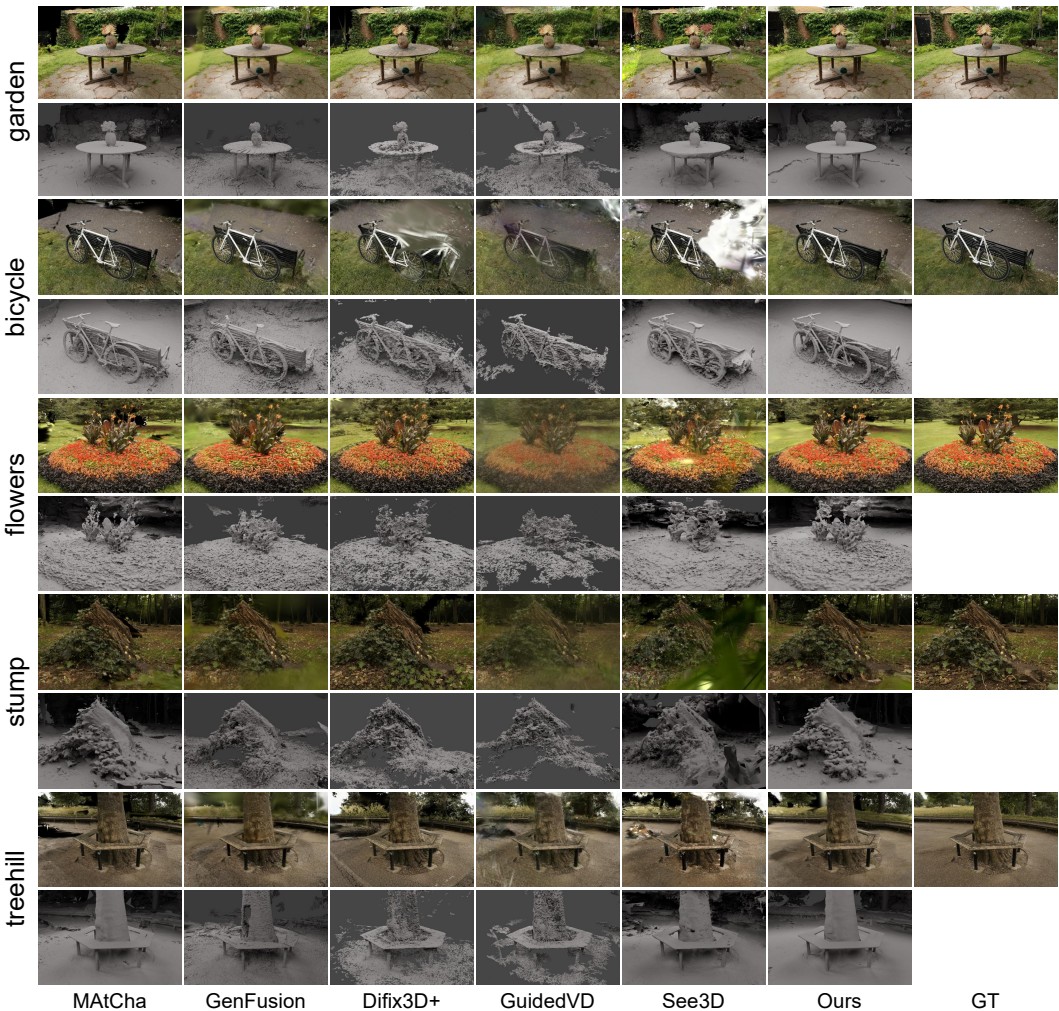

Figure A7: **Qualitative comparison using 9 input views on the Mip-NeRF 360 dataset.** We present results for all 5 outdoor scenes in the dataset. Our method produces significantly fewer Gaussian floaters and less blurriness, while recovering higher-quality and smoother geometry. These results demonstrate that our approach also performs robustly on non-planar and less structured outdoor scenes.

Table A3: **Quantitative results from 9 input views on the Mip-NeRF 360 dataset.** G4Splat performs strongly not only on indoor, Manhattan-style scenes (*bonsai, counter, kitchen, room*), but also on outdoor, non-Manhattan, and less structured scenes (*bicycle, flowers, garden, stump, treehill*), demonstrating its robustness in non-planar environments. Top-3 results are highlighted as the first , second and third .

| | bicycle | bonsai | counter | flowers | garden | kitchen | room | stump | treehill | Mean |
|---|---|---|---|---|---|---|---|---|---|---|
| **PSNR ↑** | | | | | | | | | | |
| MAtCha | 15.56 | 15.36 | 15.96 | 13.66 | 17.94 | 18.96 | 17.16 | 18.52 | 14.80 | 16.44 |
| GenFusion | 16.50 | 17.93 | 17.96 | 14.37 | 18.43 | 19.76 | 19.80 | 19.82 | 15.79 | 17.82 |
| Difix3D+ | 15.23 | 16.66 | 16.68 | 13.41 | 17.54 | 18.68 | 17.17 | 18.08 | 13.03 | 16.28 |
| GuidedVD | 14.89 | 17.21 | 17.60 | 14.23 | 18.23 | 18.00 | 18.50 | 19.00 | 13.36 | 16.78 |
| See3D | 16.08 | 16.32 | 16.49 | 15.80 | 19.94 | 19.62 | 18.23 | 18.26 | 11.51 | 16.92 |
| Ours | 18.31 | 17.97 | 19.94 | 15.48 | 19.18 | 20.84 | 20.71 | 19.53 | 15.99 | 18.66 |
| **SSIM ↑** | | | | | | | | | | |
| MAtCha | 0.281 | 0.541 | 0.549 | 0.218 | 0.390 | 0.705 | 0.616 | 0.397 | 0.381 | 0.453 |
| GenFusion | 0.296 | 0.586 | 0.577 | 0.237 | 0.339 | 0.562 | 0.667 | 0.422 | 0.383 | 0.452 |
| Difix3D+ | 0.227 | 0.551 | 0.497 | 0.185 | 0.288 | 0.530 | 0.568 | 0.339 | 0.313 | 0.389 |
| GuidedVD | 0.274 | 0.550 | 0.570 | 0.221 | 0.345 | 0.578 | 0.646 | 0.403 | 0.359 | 0.438 |
| See3D | 0.229 | 0.513 | 0.526 | 0.235 | 0.416 | 0.732 | 0.666 | 0.388 | 0.280 | 0.443 |
| Ours | 0.345 | 0.630 | 0.612 | 0.289 | 0.454 | 0.725 | 0.733 | 0.460 | 0.386 | 0.515 |
| **LPIPS ↓** | | | | | | | | | | |
| MAtCha | 0.482 | 0.362 | 0.339 | 0.520 | 0.351 | 0.241 | 0.325 | 0.441 | 0.443 | 0.389 |
| GenFusion | 0.530 | 0.384 | 0.384 | 0.568 | 0.429 | 0.335 | 0.319 | 0.493 | 0.513 | 0.439 |
| Difix3D+ | 0.509 | 0.370 | 0.378 | 0.529 | 0.408 | 0.311 | 0.350 | 0.474 | 0.501 | 0.426 |
| GuidedVD | 0.548 | 0.444 | 0.408 | 0.576 | 0.448 | 0.380 | 0.646 | 0.502 | 0.535 | 0.499 |
| See3D | 0.553 | 0.441 | 0.394 | 0.462 | 0.369 | 0.394 | 0.339 | 0.538 | 0.564 | 0.451 |
| Ours | 0.484 | 0.336 | 0.330 | 0.450 | 0.314 | 0.242 | 0.286 | 0.445 | 0.457 | 0.371 |

Table A4: **Evaluation metrics.** We show the evaluation metrics with their definitions that we use to measure reconstruction quality. $P$ and $P^*$ are the point clouds sampled from the predicted and the ground truth mesh. $n_p$ is the normal vector at point $p$.

| Metric | Definition |
|---|---|
| **Chamfer Distance (CD)** | $\frac{Accuracy+\text{Completeness}}{2}$ |
| *Accuracy* | $\underset{\boldsymbol{p}\in P}{\text{mean}}\left(\underset{\boldsymbol{p}^*\in P^*}{\min}\|\boldsymbol{p}-\boldsymbol{p}^*\|_1\right)$ |
| *Completeness* | $\underset{\boldsymbol{p}^*\in P^*}{\text{mean}}\left(\underset{\boldsymbol{p}\in P}{\min}\|\boldsymbol{p}-\boldsymbol{p}^*\|_1\right)$ |
| **F-score** | $\frac{2\times\text{Precision}\times\text{Recall}}{\text{Precision}+\text{Recall}}$ |
| *Precision* | $\underset{\boldsymbol{p}\in P}{\text{mean}}\left(\underset{\boldsymbol{p}^*\in P^*}{\min}\|\boldsymbol{p}-\boldsymbol{p}^*\|_1<0.05\right)$ |
| *Recall* | $\underset{\boldsymbol{p}^*\in P^*}{\text{mean}}\left(\underset{\boldsymbol{p}\in P}{\min}\|\boldsymbol{p}-\boldsymbol{p}^*\|_1<0.05\right)$ |
| **Normal Consistency** | $\frac{Normal\ Accuracy+Normal\ Completeness}{2}$ |
| *Normal Accuracy* | $\underset{\boldsymbol{p}\in P}{\text{mean}}\left(\boldsymbol{n}_{\boldsymbol{p}}^T\boldsymbol{n}_{\boldsymbol{p}^*}\right)$ s.t. $\boldsymbol{p}^*=\underset{\boldsymbol{p}^*\in P^*}{arg\,min}\|\boldsymbol{p}-\boldsymbol{p}^*\|_1$ |
| *Normal Completeness* | $\underset{\boldsymbol{p}^*\in P^*}{\text{mean}}\left(\boldsymbol{n}_{\boldsymbol{p}}^T\boldsymbol{n}_{\boldsymbol{p}^*}\right)$ s.t. $\boldsymbol{p}=\underset{\boldsymbol{p}\in P}{arg\,min}\|\boldsymbol{p}-\boldsymbol{p}^*\|_1$ |

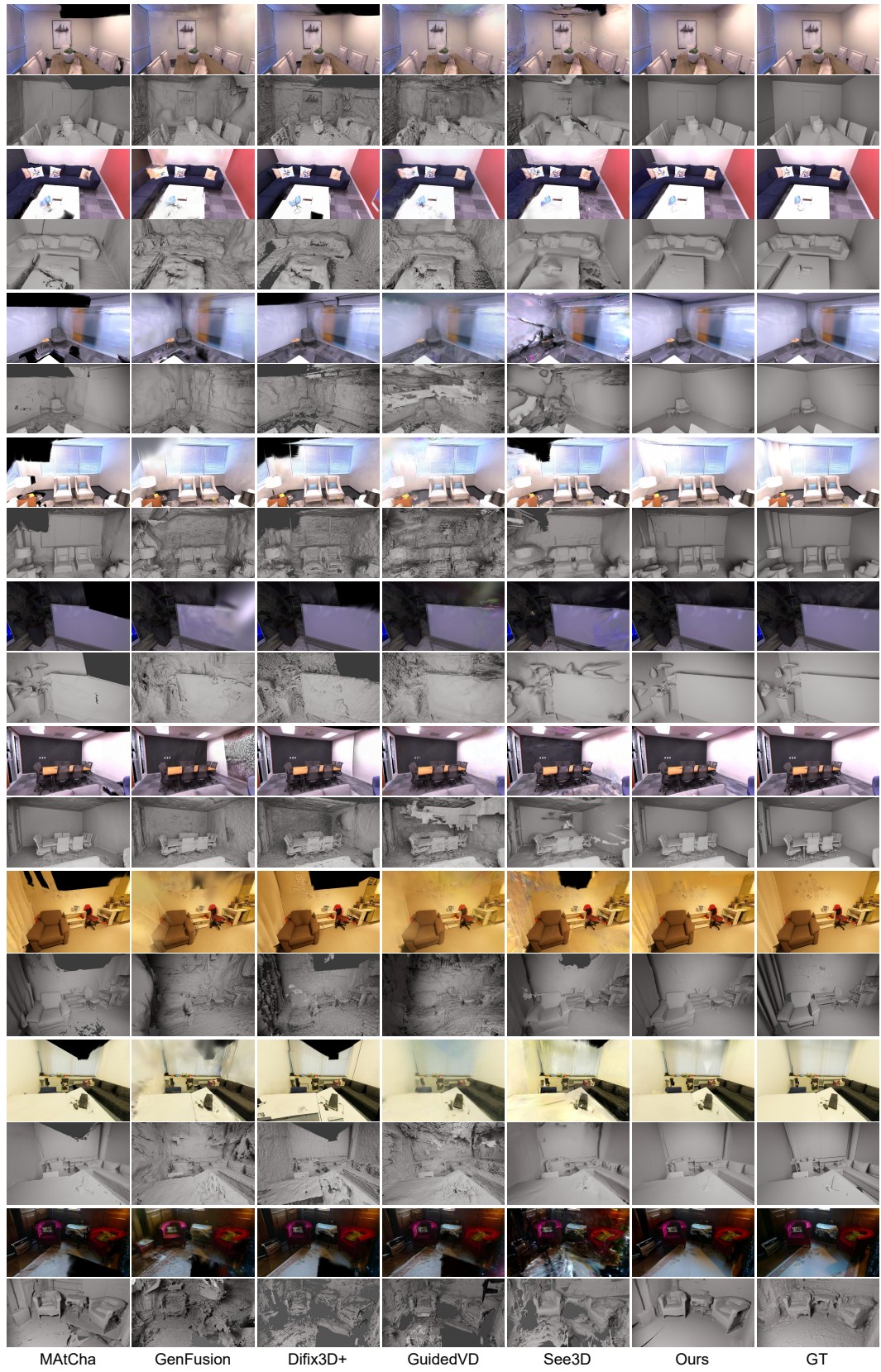

Figure A8: **More qualitative results.**

