# OpenReview forum: "G4Splat: Geometry-Guided Gaussian Splatting with Generative Prior"
_ICLR.cc/2026/Conference — ICLR 2026 Poster_

### Official Review · Reviewer_3Jys · 2025-10-31

**Soundness:** 3
**Presentation:** 3
**Contribution:** 3
**Rating:** 6
**Confidence:** 3

**Summary:**

This paper proposes G4SPLAT, a geometry-guided framework for sparse-view 3D Gaussian Splatting reconstruction. Existing generative prior methods often suffer from degraded performance due to insufficient geometric supervision and multi-view inconsistency. G4SPLAT addresses these issues by deriving scale-accurate plane-aware depth maps  that leverage the prevalent planar structures in scenes. These accurate geometric cues are integrated into a geometry-guided generation process to refine visibility mask estimation, guide novel view selection, and improve cross-view consistency during video diffusion model inpainting. Experiments on Replica, ScanNet++, and DeepBlending datasets show that G4SPLAT outperforms state-of-the-art baselines in both geometric accuracy and appearance quality.

**Strengths:**

- This paper is clearly written, with well-explained derivations of plane-aware depth maps and the geometry-guided generation process, making G4SPLAT’s contributions and implementation easy to understand.
- The proposed G4SPLAT takes advantage of plane structures in a scene to figure out depth with the right scale, which gives solid guidance even in areas that don’t have much information in sparse-view setups. Ablation results show that generative priors alone offer little geometric benefit and may even degrade quality, while adding plane-aware geometry modeling greatly improves accuracy and consistency.

**Weaknesses:**

- The proposed method relies on the presence of significant planar structures in the scene, which may not be applicable in certain complex or natural environments. In scenes lacking prominent planar features, the generated plane-aware depth maps may not provide sufficient geometric cues, potentially affecting the quality of the final 3D reconstruction.

**Questions:**

In constructing the Visibility Grid, how do the choices of voxel size and number of sampling points Q affect visibility accuracy and runtime?

---

> ### Author Response · Authors · 2025-11-21
> **Response to The Reviewer (1 / 2)**
>
> We sincerely thank the reviewer for the constructive comments on our work. We greatly appreciate your positive feedback, noting that our paper is "clearly written" and "well-explained," our method "greatly improves accuracy and consistency," and our results "outperform state-of-the-art baselines in both geometric accuracy and appearance quality." We address each of the concerns below.
>
> **1. Reliance on significant planar structures may limit applicability in complex or natural environments.**
>
> > The proposed method relies on the presence of significant planar structures in the scene, which may not be applicable in certain complex or natural environments. In scenes lacking prominent planar features, the generated plane-aware depth maps may not provide sufficient geometric cues, potentially affecting the quality of the final 3D reconstruction.
>
> We would first like to clarify that our method also performs well on non-planar or less structured scenes because it is **a strict enhancement of the base model** (e.g., 2DGS or MAtCha).
> Specifically, our method handles different types of regions as follows:
> - **Planar regions (improved)**: accuracy is enhanced by leveraging plane priors for both observed and unobserved regions;
> - **Observed non-planar regions (preserved)**: the base model’s predictions are maintained to ensure reliability;
> - **Unobserved non-planar regions (improved)**: linearly aligned monocular depth is used as a fallback where the base model provides no supervision.
>
> This design ensures that G4Splat retains the general applicability of the base model while strengthening it in planar and under-constrained areas.
>
> To further demonstrate this, we have added evaluation results on all 9 scenes from Mip-NeRF 360 using 9 input views following ReconFusion [4]. We report the averaged metrics in the table below, and provide per-scene results in Appendix Table A2, along with comparisons against baselines for all 5 outdoor scenes in Appendix Figure A5. These results show that **G4Splat performs strongly not only on indoor, Manhattan-style scenes (*bonsai, counter, kitchen, room*), but also on outdoor, non-Manhattan and less structured scenes such as *bicycle, flowers, garden, stump, and treehill*, demonstrating its robustness on non-planar environments as well.** Additionally, our original submission already included compelling examples on less-planar environments, including the *Museum* in Figure 1 and *store* and *cat* in Figure 5.
>
> | Method | PSNR↑ | SSIM↑ | LPIPS↓ |
> | -------- | -------- | -------- | -------- |
> | MAtCha     | 16.44     | 0.453     | 0.389     |
> | GenFusion     | 17.82     | 0.452     | 0.439     |
> | Difix3D+     | 16.28     | 0.389     | 0.426     |
> | GuidedVD     | 16.78     | 0.438     | 0.499     |
> | See3D     | 16.92     | 0.443     | 0.451     |
> | Ours     | **18.66**     | **0.515**     | **0.371**     |
>
> Please note that the GenFusion results we report are slightly lower than those presented in its original paper. This discrepancy arises because GenFusion initializes its point cloud using the portion of the COLMAP reconstruction generated from all available views of each scene (ranging from approximately 125 to 311 views in the dataset) that are visible to the 9 input views. Such an initialization provides an almost perfectly accurate structure in the visible regions of the input viewpoints, but it also introduces additional information beyond the 9 given views. In contrast, in our experiments, we follow the same setting as in our main paper and initialize the point cloud using MASt3R-SfM with only the 9 input views.
>
> We hope these results clearly demonstrate that **G4Splat remains robust well beyond planar or Manhattan-world settings**.
>
> [1] Neural 3D Scene Reconstruction with the Manhattan-world Assumption. CVPR 2022
>
> [2] PlanarRecon: Real-time 3D Plane Detection and Reconstruction from Posed Monocular Videos. CVPR 2022
>
> [3] IndoorGS: Geometric Cues Guided Gaussian Splatting for Indoor Scene Reconstruction. CVPR 2025
>
> [4] ReconFusion: 3D Reconstruction with Diffusion Priors. CVPR 2024
>
>
>
> **2. How do the choices of voxel size and number of sampling points Q affect visibility accuracy and runtime?**
>
> > In constructing the Visibility Grid, how do the choices of voxel size and number of sampling points Q affect visibility accuracy and runtime?
>
> In our experiments, we use a voxel resolution of 256, which is already sufficient to accurately model visibility for scenes of typical scale, and it performs well on all scenes evaluated in our paper. For exceptionally large scenes, one could increase the voxel resolution to further improve accuracy, though this would come with higher memory consumption.
>
> We define the number of sampling points along a ray, $Q$, as: $Q = \frac{d}{v_\text{min}}$
>
>
>
> where:
> - $d$ is the distance from the camera center to the surface along the ray,
> - $v_\text{min}$ is the minimal voxel size in the scene.

---

> ### Author Response · Authors · 2025-11-21
> **Response to The Reviewer (2 / 2)**
>
> This ensures that each voxel intersected by the ray contains at least one sample, which ensures accurate visibility estimation.
>
> Moreover, because our method computes the visibility value of each sampled point via trilinear interpolation implemented in PyTorch, the computation is highly efficient and benefits directly from CUDA parallelization.

---

> > ### Comment · Reviewer_3Jys · 2025-11-27
> >
> > Thank you for the thorough response. I appreciate the clarification of how G4SPLAT improves both planar and non-planar regions, the visual results in the paper already look quite appealing. I will keep my original score of 6. Wish you all the best with the work.

---

### Official Review · Reviewer_jjWR · 2025-10-31

**Soundness:** 3
**Presentation:** 4
**Contribution:** 3
**Rating:** 6
**Confidence:** 4

**Summary:**

The paper introduces G4Splat, a new method for sparse-view 3D scene reconstruction. Building upon existing work MAtCha Gaussians [1], the paper proposes to leverage estimated 3D planes throughout the entire reconstruction pipeline that additionally leverages a pre-trained video diffusion model as a prior. Making use of the usual existence of planar structures in 3D scenes created by humans, the authors propose to extract per-view 2D planes with SAM in order to merge them into global 3D planes.
These can then be used to obtain more accurate depth than from sparse-view SfM approaches like MASt3R-SfM [2] used in [1], and to compute improved visibility masks compared to rendered alpha maps, which are required as inpainting masks when combined with a generative prior.
The authors further leverage the 3D planes for a plane-aware selection of novel views for inpainting and finally for selecting supervision signals from the geometry-guided inpainting.
An experimental evaluation on ScanNet++, DeepBlending, and Replica shows that G4Splat consistently outperforms all baselines w.r.t. both 3D surface reconstruction and novel view synthesis quality. The paper further includes qualitative results for different numbers of input views and an ablation study w.r.t. the use of the generative prior, the 3D planes, and both combined.

- [1] MAtCha Gaussians: Atlas of Charts for High-Quality Geometry and Photorealism From Sparse Views. CVPR 2025
- [2] MASt3R-SfM: a Fully-Integrated Solution for Unconstrained Structure-from-Motion. 3DV 2025

**Strengths:**

- The paper is well written and mostly easy to follow and understand.
  - The introduction motivates the topic well and discusses the idea following the Manhattan world assumption and the shortcomings of existing works leveraging generative priors.
  - The related work section is comprehensive. I found Sec. 2.3. about the plane assumption in reconstruction especially interesting.
  - The method section first introduces the 3D plane estimation step by step (first 2D and per view, then 3D and global), followed by all applications of the obtained 3D planes throughout the entire optimization pipeline.
- I appreciate the method and paper story of making use of 3D planes everywhere throughout the 3D reconstruction pipeline, following the Manhattan world assumption.
  - The step-wise estimation of the 3D planes using SAM in 2D, then merging these with RANSAC, is a reasonable and intuitive approach.
  - The paper shows how to obtain more accurate depth maps and visibility masks using 3D planes as well as more useful novel view selection for inpainting with generative priors.
    - Fig. 3 visualizes all of these improvements over naive alternatives well.
- The quantitative and qualitative comparisons with baselines are overall very convincing.
  - The proposed method consistently and significantly outperforms all baselines for both 3D surface reconstruction and novel view synthesis.
  - The authors make sure to include a baseline of 2DGS combined with the video diffusion model that they use (See3D), as this model seems to be quite new and not used by baselines.
  - The authors choose very recent and strong baselines, representing the current state of the art.
- The runtime comparison shows that the proposed method is roughly as fast as previous works leveraging generative priors, while obtaining better quality.
- The ablation study shows that all individual design choices contribute to the overall performance of the method.
- The authors provide additional convincing qualitative results in form of videos on the anonymous website.
- The appendix provides more results, detailed background to prior works that this paper leverages, training details, and limitations.

**Weaknesses:**

- Since the method relies on 3D planes, it seems to be quite tailored for indoor and possibly city-like outdoor scenes, or in other words, scenes that actually consist of enough 3D planes.
  - I am wondering how this method would perform on other kind of outdoor scenes, e.g., the garden or bike scenes in the MipNeRF360 dataset. Are there still enough planar structures to leverage?
    - In the limitations section in the appendix, the authors do touch on this topic but mainly for obtaining scale-accurate depth.
  - The paper, appendix, and website mostly show almost only indoor scenes.
- The paper would benefit from a more precise evaluation of 3D reconstruction for observed and unobserved regions separately.
  - It would be very interesting to see quantitatively how much each component improves observed and unobserved regions, respectively.
- Concern about fairness w.r.t. generative prior:
  - While the authors do provide a baseline of 2DGS + See3D, i.e., the video generative prior that they use in this paper, would MAtCha + See3D not be the better baseline in terms of fairness?
    - In Fig.4, MAtCha looks quite accurate, just incomplete, whereas 2DGS + See3D has a lot of artifacts. Is this solely due to the generative prior See3D or also due to 2DGS being worse than MAtCha?
  - Regarding the use of generative priors, the main paper (related work) would need some more details about which approach uses what kind of generative prior and how does it affect results.
  - To this end, it would be beneficial for the paper to evaluate their method with the generative prior used by baselines, if that is possible. This would be important in order to precisely attribute the performance gains to a better generative prior or to the other technical contributions, e.g., using the 3D planes.
- Some lack of clarity:
  - The need for the background Sec. 3.1 about MAtCha Gaussians became only clear much later in Sec. 3.4 that then states that G4Splat builds on MAtCha as initialization. There is no description of the structure of Sec. 3 (after the Method title), but the paper just goes immediately into the background section after the related work, leaving the reader confused why another "related work" section in the method section is needed. Writing one or two sentences about the structure of the Method section and the individual subsections and why they are necessary at the beginning of the Method section would resolve this.
  - From the main paper, it is unclear what role 3D plane extrapolation plays in the pipeline. Are 3D planes extrapolated to unobserved regions? Does this involve any additional assumptions about the layout of an indoor scene, for example?
    - Line 74f.: "planar surfaces allow depth extrapolation: a 3D plane can be reliably estimated from partial depth observations and then extended across the entire surface" states something along those lines.
  - In paragraph "Per-view 2D Plane Extraction" (lines 193ff.), it is unclear how you obtain the normals.
  - Is the depth from the monocular depth estimator and scaled using the 3D plane geometry really better than the depth estimated by MASt3R-SfM? If this is the case, why? Are more recent DUSt3R follow-ups like VGGT maybe more performant?
  - The color supervision description in lines 314f. is quite vague and refers to the appendix. If possible, it would be beneficial to move this information to the main paper to make it more self-contained, as otherwise there remain open questions that require the appendix.
  - Tab.3 misses to explain what "DS" in "Ours (DS)" stands for. Actually, I was not able to find that information anywhere in the paper. I assume it uses a distilled video model maybe?
  - The PM setting in the ablation study is not completely clear to me. Are the planes there only used for initialization and for depth supervision or for what parts of the pipeline exactly?
- The paper misses a related work and potential baseline: Spurfies [1].


References:
- [1] Spurfies: Sparse Surface Reconstruction using Local Geometry Priors. 3DV 2025

**Questions:**

My suggestions to the authors are already detailed in the weaknesses section but here again concretely:
- It would be interesting to evaluate the approach on more outdoor scenes with less planar structures like the garden and bike scenes in the MipNeRF360 dataset.
  - While I do see the value in a method that performs particularly well for indoor scenes, this would be beneficial for the paper to better understand possible limitations.
- Splitting the quantitative (and possibly qualitative) evaluation into observed and unobserved regions would be very interesting to evaluate the behavior of the individual contributions more precisely.
- I suggest that the authors evaluate MAtCha + See3D (or is that already the second row, i.e., GP only of the ablation study maybe) instead of 2DGS + See3D. It would be the better comparison as a baseline in qualitative and quantitative evaluations in terms of fairness.
- It would be interesting to also evaluate G4Splat in combination with different generative priors to see whether the 3D planes can make the use of generative priors more effective irrespective of the particular choice which prior to use.
- Further open questions are:
  - Are 3D planes extrapolated to unobserved regions? Does this involve additional assumptions? If so, how does it affect results?
  - How are surface normals obtained (cf. lines 193ff.)?
  - Why is the depth from monocular depth estimation and scaling better than from dense reconstruction approaches like MASt3R-SfM or VGGT?
  - What does "DS" stand for in Tab.3?
  - Could you give details about the PM setting in the ablation?

---

> ### Author Response · Authors · 2025-11-21
> **Response to The Reviewer (1 / 4)**
>
> We sincerely thank the reviewer for the constructive comments on our work. We greatly appreciate your positive feedback: you found our paper "well written and mostly easy to follow and understand," our method "reasonable and intuitive," and our results "overall very convincing" and "outperforming all baselines w.r.t both 3D surface reconstruction and novel view synthesis quality." More importantly, we are especially grateful for your valuable suggestions to further improve our work, including the evaluation on observed/unobserved regions, testing with different generative models, and enhancing writing clarity. We address each of the concerns below.
>
> **1. Evaluate the approach on more outdoor scenes with less planar structures.**
>
> > It would be interesting to evaluate the approach on more outdoor scenes with less planar structures like the garden and bike scenes in the MipNeRF360 dataset.
>
> Thanks for your suggestion to evaluate our method on less planar scenes such as the garden and bike scenes from the Mip-NeRF 360 dataset. We have conducted the corresponding experiments and will present the results shortly. Before that, we would like to clarify that our method also performs well on non-planar or less structured scenes because it is **a strict enhancement of the base model** (e.g., 2DGS or MAtCha).
> Specifically, our method handles different types of regions as follows:
> - **Planar regions (improved)**: accuracy is enhanced by leveraging plane priors for both observed and unobserved regions;
> - **Observed non-planar regions (preserved)**: the base model’s predictions are maintained to ensure reliability;
> - **Unobserved non-planar regions (improved)**: linearly aligned monocular depth is used as a fallback where the base model provides no supervision.
>
> This design ensures that G4Splat retains the general applicability of the base model while strengthening it in planar and under-constrained areas.
>
> To further demonstrate this, we have added evaluation results on all 9 scenes from Mip-NeRF 360 using 9 input views following ReconFusion [4]. We report the averaged metrics in the table below, and provide per-scene results in Appendix Table A2, along with comparisons against baselines for **all 5 outdoor scenes in Appendix Figure A5**. These results show that G4Splat **performs strongly not only on indoor, Manhattan-style scenes** (*bonsai, counter, kitchen, room*), **but also on outdoor, non-Manhattan and less structured scenes** such as *bicycle, flowers, garden, stump, and treehill*, demonstrating its robustness on non-planar environments as well.
>
> | Method | PSNR↑ | SSIM↑ | LPIPS↓ |
> | -------- | -------- | -------- | -------- |
> | MAtCha     | 16.44     | 0.453     | 0.389     |
> | GenFusion     | 17.82     | 0.452     | 0.439     |
> | Difix3D+     | 16.28     | 0.389     | 0.426     |
> | GuidedVD     | 16.78     | 0.438     | 0.499     |
> | See3D     | 16.92     | 0.443     | 0.451     |
> | Ours     | **18.66**     | **0.515**     | **0.371**     |
>
> Please note that the GenFusion results we report are slightly lower than those presented in its original paper. This discrepancy arises because GenFusion initializes its point cloud using the portion of the COLMAP reconstruction generated from all available views of each scene (ranging from approximately 125 to 311 views in the dataset) that are visible to the 9 input views. Such an initialization provides an almost perfectly accurate structure in the visible regions of the input viewpoints, but it also introduces additional information beyond the 9 given views. In contrast, in our experiments, we follow the same setting as in our main paper and initialize the point cloud using MASt3R-SfM with only the 9 input views.
>
> We hope these results clearly demonstrate that **G4Splat remains robust well beyond planar or Manhattan-world settings**.
>
> [1] ReconFusion: 3D Reconstruction with Diffusion Priors. CVPR 2024
>
>
> **2. Splitting the quantitative (and possibly qualitative) evaluation into observed and unobserved regions.**
>
> > Splitting the quantitative (and possibly qualitative) evaluation into observed and unobserved regions would be very interesting to evaluate the behavior of the individual contributions more precisely.
>
>
> Thank you for the suggestion. We have evaluated all methods separately on observed and unobserved regions, and the results are shown in the table below. We also include a **visualization of the visibility mask in Appendix Figure A6** (in the GT images, the golden regions indicate unobserved areas across the 5 input views).

---

> ### Author Response · Authors · 2025-11-21
> **Response to The Reviewer (2 / 4)**
>
> | Method | CD↓ (Observed Region) | PSNR↑ (Observed Region) | CD↓ (Unobserved Region) | PSNR↑ (Unobserved Region) |
> | -------- | -------- | -------- | -------- | -------- |
> | MAtCha     | 5.73     | 19.70     | 30.58     | 17.23     |
> | GenFusion     | 10.34     | 20.75     | 23.60     | 19.53     |
> | Difix3D+ | 9.13 | 20.94 | 33.60 | 18.57 |
> | GuidedVD |  18.28 | 23.03 | 47.26 | 22.14 |
> | Ours | **5.45** | **24.71** | **10.16** | **23.33** |
>
> Evaluating observed and unobserved regions separately **clearly demonstrates** that G4Splat not only improves performance in the regions seen by the input views compared to the baselines, but also achieves more significant improvements in unobserved regions, producing **smoother and more accurate reconstructions with fewer Gaussian floaters**. We have added this experiment to Sec. 4.2 of the revised paper to further validate the effectiveness of our method.
>
>
> **3. Evaluation of MAtCha + See3D as a baseline in qualitative and quantitative comparisons.**
>
> > I suggest that the authors evaluate MAtCha + See3D (or is that already the second row, i.e., GP only of the ablation study maybe) instead of 2DGS + See3D. It would be the better comparison as a baseline in qualitative and quantitative evaluations in terms of fairness.
>
> The "GP only" row in Table 2 of our ablation study indeed **corresponds to MAtCha + See3D**, and we have compiled and reported the results in the table below. Moreover, we have included a **visualization in appendix Figure A7** comparing MAtCha + See3D with other methods. From the results, it is clear that incorporating the See3D diffusion prior on top of MAtCha yields better performance than applying it on 2DGS. This improvement primarily comes from MAtCha’s scale-accurate depth supervision. Although this depth is less accurate than the plane-aware depth used in our method, it is still considerably more reliable than the depth distortion loss employed in 2DGS.
>
> | Method | CD↓ | F-Score↑ | NC↑ | PSNR↑ | SSIM↑ | LPIPS↓ |
> | -------- | -------- | -------- | -------- | -------- | -------- | -------- |
> | MAtCha     | 10.12     | 60.90     | 79.33     | 17.81     | 0.752     | 0.228     |
> | See3D (w. 2DGS)     | 12.74     | 45.27     | 73.98     | 19.22     | 0.735     | 0.328     |
> | See3D (w. MAtCha)     | 9.46     | 56.99     | 77.58     | 19.63     | 0.740     | 0.295     |
> | Ours (w. See3D)     | **6.61**     | **65.14**     | **83.98**     | **23.90**     | **0.836**     | **0.199**     |
>
>
>
> **4. Evaluate G4Splat in combination with different generative priors.**
>
> > It would be interesting to also evaluate G4Splat in combination with different generative priors to see whether the 3D planes can make the use of generative priors more effective irrespective of the particular choice which prior to use.
>
> Thank you for the suggestion. We conducted an additional experiment where we replaced the See3D diffusion model with ViewCrafter, which is used in GuidedVD, and evaluated G4Splat on all 8 Replica scenes from 5 input views. The quantitative results are shown in the table below, and we also include **visual comparisons in appendix Figure A8**.
>
> | Method | CD↓ | F-Score↑ | NC↑ | PSNR↑ | SSIM↑ | LPIPS↓ |
> | -------- | -------- | -------- | -------- | -------- | -------- | -------- |
> | MAtCha     | 10.12     | 60.90     | 79.33     | 17.81     | 0.752     | 0.228     |
> | GenFusion     | 13.05     | 41.60     |  69.33    | 20.14     | 0.801     | 0.258     |
> | Difix3D+     | 13.71     | 43.11     | 65.34     | 19.42     | 0.779     | 0.231     |
> | GuidedVD     | 27.87     | 17.29     | 61.64     | 22.51     | 0.822     | 0.260     |
> | Ours (w. ViewCrafter)     | 7.95     | 64.19     | 81.82     | 22.08     | 0.812     | 0.210     |
> | Ours (w. See3D)     | **6.61**     | **65.14**     | **83.98**     | **23.90**     | **0.836**     | **0.199**     |
>
> From both the quantitative and qualitative results, we observe that G4Splat maintains **high-quality and smooth geometry reconstruction** even when using a weaker generative prior model. All reconstruction metrics remain superior to the baselines. In terms of rendering quality, G4Splat continues to effectively **suppress Gaussian floaters and blurriness**, achieving strong performance on the LPIPS metric as well.
>
> This experiment demonstrates that our method is compatible with different generative diffusion models, consistently achieving strong performance across them. We have added this experiment to the ablation study in Sec. 4.3 of the revised paper.

---

> ### Author Response · Authors · 2025-11-21
> **Response to The Reviewer (3 / 4)**
>
> **5. Are 3D planes extrapolated to unobserved regions? Does this involve additional assumptions? If so, how does it affect results?**
>
> > Are 3D planes extrapolated to unobserved regions? Does this involve additional assumptions? If so, how does it affect results?
>
> Yes, our method also extrapolates 3D planes to unobserved regions, and this **does not require introducing any additional assumptions**. In fact, this design is a key reason why G4Splat achieves significantly better performance than the baselines in unobserved areas.
>
> More specifically, after inpainting novel views using See3D, we include these views in the training view set. They then participate jointly with the input views in our plane-aware geometry modeling (as described in Sec. 3.2 and Figure 3 of the paper). In particular, the global 3D plane estimation step **merges per-view 2D planes belonging to the same global plane across all training views**. This naturally extrapolates the 3D planes to the unobserved planar regions of the novel views, without requiring any extra assumptions.
>
>
> **6. How are surface normals obtained (cf. lines 193ff.)?**
>
> > How are surface normals obtained (cf. lines 193ff.)?
>
> In our implementation, for simplicity, we obtain the surface normals by **computing the gradients of the depth maps**, following the same procedure used in MAtCha. This generates a normal map directly from the depth geometry. Of course, we could also use a **monocular normal estimator** such as StableNormal [1] to predict surface normals, and our framework is compatible with that option as well. We have added this clarification to Sec. 3.2 of the main paper in the revised version to avoid confusion.
>
> [1] StableNormal: Reducing Diffusion Variance for Stable and Sharp Normal. SIGGRAPH Asia 2024
>
>
> **7. Why is the depth from monocular depth estimation and scaling better than from dense reconstruction approaches like MASt3R-SfM or VGGT?**
>
> > Why is the depth from monocular depth estimation and scaling better than from dense reconstruction approaches like MASt3R-SfM or VGGT?
>
>
> As discussed above, our method is a strict enhancement of the base model (e.g., 2DGS or MAtCha) and treats different types of regions as follows:
> - **Planar regions (improved)**: accuracy is enhanced by leveraging plane priors for both observed and unobserved regions;
> - **Observed non-planar regions (preserved)**: the base model’s predictions are maintained to ensure reliability;
> - **Unobserved non-planar regions (improved)**: linearly aligned monocular depth is used as a fallback where the base model provides no supervision.
>
> Crucially, monocular depth (after linear scaling) is used only in these unobserved non-planar regions, where neither the base model (MAtCha in our implementation, but could similarly be MASt3R-SfM or VGGT) nor the plane priors offer guidance. In other words, monocular depth never replaces dense reconstruction where reliable estimates already exist; it is applied solely as a fallback mechanism.
>
> If any points remain unclear, we would be happy to further discuss them during the discussion stage.
>
>
> **8. What does "DS" stand for in Tab.3?**
>
> > Tab.3 misses to explain what "DS" in "Ours (DS)" stands for. Actually, I was not able to find that information anywhere in the paper. I assume it uses a distilled video model maybe? What does "DS" stand for in Tab.3?
>
> Thank you for pointing this out. DS refers to our accelerated variant that downsamples the initial Gaussians (DS is short for DownSample). This variant substantially reduces runtime while still outperforming all baselines. In the original submission, DS was described only in Appendix C.7 (Implementation Details). We have moved this clarification to Sec 4.4 of main paper in the revised version to avoid confusion.
>
>
>
> **9. Could you give details about the PM setting in the ablation?**
>
> > The PM setting in the ablation study is not completely clear to me. Are the planes there only used for initialization and for depth supervision or for what parts of the pipeline exactly? Could you give details about the PM setting in the ablation?
>
> In our ablation study, enabling plane-aware geometry modeling (PM) means that we compute plane-aware depth maps, and these maps are then used for both Gaussian initialization and depth supervision.
>
>
> **10. The paper misses a related work and potential baseline: Spurfies.**
>
> > The paper misses a related work and potential baseline: Spurfies
>
> Thank you for pointing this out. We have included Spurfies in the Sec 2.2 of the revised paper. In addition, since the MAtCha paper reports that MAtCha outperforms Spurfies, and our experimental results show that our method further outperforms MAtCha, it is reasonable to conclude that our method would also surpass Spurfies.

---

> ### Author Response · Authors · 2025-11-21
> **Response to The Reviewer (4 / 4)**
>
> **11. Writting suggestions.**
>
> > The need for the background Sec. 3.1 about MAtCha Gaussians became only clear much later in Sec. 3.4 that then states that G4Splat builds on MAtCha as initialization. There is no description of the structure of Sec. 3 (after the Method title), but the paper just goes immediately into the background section after the related work, leaving the reader confused why another "related work" section in the method section is needed. Writing one or two sentences about the structure of the Method section and the individual subsections and why they are necessary at the beginning of the Method section would resolve this.
>
> > The color supervision description in lines 314f. is quite vague and refers to the appendix. If possible, it would be beneficial to move this information to the main paper to make it more self-contained, as otherwise there remain open questions that require the appendix.
>
> Thank you very much for these helpful writing suggestions. We have revised the paper accordingly to improve clarity and self-containment.

---

> > ### Author Response · Authors · 2025-11-28
> >
> > Thank you very much for your thoughtful comments and the time you have invested in reviewing our paper. We have provided a detailed response to the concerns you raised earlier. Could you please let us know if our clarification sufficiently addressed your questions? If anything remains unclear, we would be more than happy to continue the discussion.

---

### Official Review · Reviewer_HhvE · 2025-11-01

**Soundness:** 3
**Presentation:** 3
**Contribution:** 2
**Rating:** 4
**Confidence:** 3

**Summary:**

G4Splat proposes a reconstruction pipeline for sparse-view 3D scene reconstruction that integrates metric-scale geometry estimation (leveraging planar priors) with generative priors from pretrained diffusion/image models. Geometry guidance is injected at multiple stages (depth estimation, visibility masks, view selection, and inpainting) to reduce shape–appearance ambiguities and improve multi-view consistency in novel-view completion. Experiments on Replica, ScanNet++ and DeepBlending report improvements in both geometry and appearance, especially in unobserved regions.

**Strengths:**

Identifies a real limitation in prior generative-prior reconstructions (poor geometry in observed areas and inconsistency in unobserved areas) and supplies a concrete fix via planar metric depth and guided inpainting.

End-to-end incorporation of geometry at multiple pipeline points (visibility masks, view selection) is sensible and likely to increase multi-view consistency.

Broad evaluation on several standard datasets and claimed improvements on both geometry and appearance metrics.

**Weaknesses:**

Reliance on planar priors: in scenes without significant planar structure (e.g., natural outdoor scenes, complex organic interiors), the metric-depth derivation may fail; robustness experiments for such cases are not prominent in the material on the forum page.

Integration with generative priors can still propagate biases from the generative model (style/appearance biases) - the paper does not analyze or mitigate such biases.

Computational cost: combining geometry estimation, diffusion-based inpainting, and splatting can be expensive; readers would benefit from runtime and resource-use breakdowns and ablations on where the gains come from.

**Questions:**

Provide quantitative robustness experiments on scenes with few planar structures — how does the depth prior behave and how does it affect final reconstructions?

How does G4Splat handle scale ambiguity when planar cues are incorrect or scarce? Provide failure cases.

Please report computational cost and latency for a representative scene, and ablate which component (plane-based depth, guided visibility, diffusion inpainting) provides the largest gain.

---

> ### Author Response · Authors · 2025-11-21
> **Response to The Reviewer (1 / 2)**
>
> We sincerely thank the reviewer for the constructive comments on our work. We greatly appreciate your positive feedback, particularly your recognition that our paper "identifies a real limitation in prior generative-prior reconstructions," that our method "is sensible and likely to increase multi-view consistency," and that our results demonstrate "improvements in both geometry and appearance, especially in unobserved regions." Regarding some of the questions you raised, we would like to note respectfully that several of these points were already analyzed in the main paper, including the discussion of generative bias in Sec. 3.3, the computational cost in Table 3, and the ablation studies in Table 2 and Sec. 4.3. We hope this clarification is helpful, and We address each of the concerns below.
>
> **1. Reliance on planar priors (need more robustness experiments on scenes with few planar structures).**
>
>
> > Reliance on planar priors: in scenes without significant planar structure (e.g., natural outdoor scenes, complex organic interiors), the metric-depth derivation may fail; robustness experiments for such cases are not prominent in the material on the forum page.
>
> We would first like to clarify that our method also performs well on non-planar or less structured scenes because it is **a strict enhancement of the base model** (e.g., 2DGS or MAtCha).
> Specifically, our method handles different types of regions as follows:
> - **Planar regions (improved)**: accuracy is enhanced by leveraging plane priors for both observed and unobserved regions;
> - **Observed non-planar regions (preserved)**: the base model’s predictions are maintained to ensure reliability;
> - **Unobserved non-planar regions (improved)**: linearly aligned monocular depth is used as a fallback where the base model provides no supervision.
>
> This design ensures that G4Splat retains the general applicability of the base model while strengthening it in planar and under-constrained areas.
>
> To further demonstrate this, we have added evaluation results on all 9 scenes from Mip-NeRF 360 using 9 input views following ReconFusion [4]. We report the averaged metrics in the table below, and provide per-scene results in Appendix Table A2, along with comparisons against baselines for **all 5 outdoor scenes in Appendix Figure A5**. These results show that G4Splat **performs strongly not only on indoor, Manhattan-style scenes** (*bonsai, counter, kitchen, room*), **but also on outdoor, non-Manhattan and less structured scenes** such as *bicycle, flowers, garden, stump, and treehill*, demonstrating its robustness on non-planar environments as well. Additionally, our original submission already included compelling examples on less-planar environments, including the *Museum* in Figure 1 and *store* and *cat* in Figure 5.
>
> | Method | PSNR↑ | SSIM↑ | LPIPS↓ |
> | -------- | -------- | -------- | -------- |
> | MAtCha     | 16.44     | 0.453     | 0.389     |
> | GenFusion     | 17.82     | 0.452     | 0.439     |
> | Difix3D+     | 16.28     | 0.389     | 0.426     |
> | GuidedVD     | 16.78     | 0.438     | 0.499     |
> | See3D     | 16.92     | 0.443     | 0.451     |
> | Ours     | **18.66**     | **0.515**     | **0.371**     |
>
> Please note that the GenFusion results we report are slightly lower than those presented in its original paper. This discrepancy arises because GenFusion initializes its point cloud using the portion of the COLMAP reconstruction generated from all available views of each scene (ranging from approximately 125 to 311 views in the dataset) that are visible to the 9 input views. Such an initialization provides an almost perfectly accurate structure in the visible regions of the input viewpoints, but it also introduces additional information beyond the 9 given views. In contrast, in our experiments, we follow the same setting as in our main paper and initialize the point cloud using MASt3R-SfM with only the 9 input views.
>
> We hope these results clearly demonstrate that **G4Splat remains robust well beyond planar or Manhattan-world settings**.
>
> [1] Neural 3D Scene Reconstruction with the Manhattan-world Assumption. CVPR 2022
>
> [2] PlanarRecon: Real-time 3D Plane Detection and Reconstruction from Posed Monocular Videos. CVPR 2022
>
> [3] IndoorGS: Geometric Cues Guided Gaussian Splatting for Indoor Scene Reconstruction. CVPR 2025
>
> [4] ReconFusion: 3D Reconstruction with Diffusion Priors. CVPR 2024

---

> ### Author Response · Authors · 2025-11-21
> **Response to The Reviewer (2 / 2)**
>
> **2. Biases from integration with generative priors.**
>
> > Integration with generative priors can still propagate biases from the generative model (style/appearance biases) - the paper does not analyze or mitigate such biases.
>
> In fact, **our original submission already provides a detailed analysis and mitigation strategy** for the potential biases introduced by generative priors. First, the geometry-guided generative pipeline proposed in Sec. 3.3, which includes geometry-guided visibility, plane-aware novel-view selection, and geometry-guided inpainting, is specifically **designed to reduce inconsistency and bias when integrating generative priors** by enforcing accurate geometric constraints throughout the refinement process.
>
> Furthermore, **Appendix Sec. C.3 and Figure A3 present a comprehensive comparison** between directly applying generative priors (which indeed leads to biased or inconsistent results) and applying our geometry-guided generative pipeline, which significantly alleviates these issues. Finally, we also provide **representative failure cases in Appendix Sec. D**, illustrating the remaining limitations that arise from generative-prior biases.
>
> We hope this clarifies that the paper not only acknowledges the issue but also analyzes and mitigates it through accurate geometric guidance.
>
>
> **3. Computational cost and runtime comparison.**
>
> > Please report computational cost and latency for a representative scene
>
> In fact, **our original submission already reports a running time comparison** with all baselines in Table 3 of the main paper. Reviewer jjWR also noted in the comment that *"The runtime comparison shows that the proposed method is roughly as fast as previous works leveraging generative priors, while obtaining better quality."*
>
> As shown in Table 3, our method achieves the best reconstruction quality while being as fast as previous works that leverage generative priors. Moreover, our accelerated variant *Ours (DS)* (which downsamples the initial Gaussians) significantly reduces runtime while still outperforming all baselines.
>
>
> **4. Ablation study on which component in our method provides the largest gain.**
>
> > ablate which component (plane-based depth, guided visibility, diffusion inpainting) provides the largest gain.
>
> In fact, **our original submission already includes a detailed ablation study on all these components** in Sec. 4.3 and Table 2 of the main paper. This study covers diffusion inpainting (i.e., generative prior, GP), plane-based depth (i.e., plane-aware geometry modeling, PM), and guided visibility (i.e., geometry-guided generative pipeline, PP). Please refer to the corresponding sections in the main paper for a thorough analysis of the contributions of each component.
>
> Additionally, both Reviewer 56rG (*"ablations show benefit over raw monocular depth"*) and Reviewer jjWR (*"The ablation study shows that all individual design choices contribute to the overall performance of the method"*) ackownledge the comprehensiveness and clarity of our ablation experiments.
>
>
> **5. How does G4Splat handle scale ambiguity when planar cues are incorrect or scarce?**
>
> > How does G4Splat handle scale ambiguity when planar cues are incorrect or scarce? Provide failure cases.
>
> As discussed above and demonstrated on the Mip-NeRF 360 dataset, G4Splat achieves performance superior to all baselines even in scenes where planar cues are scarce. A detailed analysis of this setting is provided earlier in our response.
>
> In addition, G4Splat is inherently **robust and can correct inaccurate planar cues** during iterative optimization. In most cases, incorrect planar estimates arise because too few confident points belong to the underlying plane, leading to errors in the RANSAC fitting. During optimization, however, G4Splat progressively merges multiple local per-view planes that correspond to the same global planar surface and then refits the single plane using the confident points aggregated from all merged local planes. This consolidation process effectively corrects local planes that may initially be inaccurate. For further details, please refer to Sec. 3.2 of the main paper.

---

> > ### Author Response · Authors · 2025-11-28
> >
> > Thank you very much for your thoughtful comments and the time you have invested in reviewing our paper. We have provided a detailed response to the concerns you raised earlier. Could you please let us know if our clarification sufficiently addressed your questions? If anything remains unclear, we would be more than happy to continue the discussion.

---

### Official Review · Reviewer_56rG · 2025-11-01

**Soundness:** 2
**Presentation:** 2
**Contribution:** 1
**Rating:** 4
**Confidence:** 5

**Summary:**

This paper proposes G4SPLAT, a sparse-view 3DGS reconstruction method that uses planar depth priors and geometry-guided video diffusion refinement. The idea is to extract global 3D planar surfaces from input images, convert them into scale-accurate depth maps, and use these to supervise Gaussian optimization, select novel viewpoints, and constrain generative inpainting. The method iteratively refines the 3D model using inpainted novel views and recomputed plane-aware depth maps. Experiments on Replica, ScanNet++, and DeepBlending claim substantial gains in unobserved regions and single-view/unposed settings. Overall, the pipeline combines MAtCha-style plane-scaled depth, 2DGS, and video diffusion priors, but introduces significant engineering complexity in return for modest conceptual novelty.

**Strengths:**

- Clear motivation: improving geometry for generative 3DGS. The paper correctly identifies geometry accuracy as a limiting factor in generative scene completion and explicitly seeks to address shape-appearance ambiguity. This motivation is grounded in observed weaknesses of recent diffusion-enhanced 3DGS methods.
- Plane-based scale recovery is well-explained and technically coherent. The method extends plane fits across views using SAM, normal clustering, and RANSAC with multi-view consistency checks, yielding scale-aligned depth even in weakly observed areas. While not new, the pipeline is carefully engineered, and ablations show benefit over raw monocular depth.
- The approach uses plane-aware visibility masks, plane-driven novel-view planning, and plane-based inpainting weighting to enforce consistency with the goal of stabilizing diffusion-assisted NVS. This produces fewer floaters and sharper planar regions compared to GenFusion / Difix3D+.

**Weaknesses:**

- **Fundamental Reliance on Planar Structures:** The method's core contribution and primary advantage are fundamentally tied to the Manhattan-world assumption. While effective for artificial environments, this reliance makes the approach far less suitable for organic, non-planar scenes (e.g., natural landscapes, complex statues, foliage). The paper's solution for non-planar regions is to fall back on monocular depth estimation, which is the very technique it criticizes for scale ambiguity.
- **Heavy Engineering Pipeline** - G4Splat is not a single model but a complex, multi-stage pipeline that glues together numerous off-the-shelf components like MAtCha, MASt3R-SfM, SAM, K-means clustering, RANSAC, a monocular depth estimator, and a video diffusion model. This high complexity makes the system brittle - a failure in any one component could compromise the entire pipeline.
- The approach inherits a strong geometric scaffold from MAtCha, including scale-aligned depth, plane priors, and reliable surface initialization. This prior stabilizes 3DGS optimization in sparse-view regimes and likely contributes to the improved geometric integrity. In contrast, several baselines (e.g., GenFusion and Difix3D+) do not assume a comparable geometric initialization and instead operate in a more challenging setting where depth must be inferred solely from generative consistency. This makes the comparison somewhat imbalanced, and it becomes difficult to isolate how much of the improvement stems from the proposed refinements versus MAtCha's initialization advantage.
- The qualitative comparisons in Fig. 4 appear to show holes and structural failures for generative baselines that are not typically reported in their original papers. This behavior is plausible when such models are applied without a metric depth prior or explicit plane constraints, especially under sparse or weakly posed conditions. However, the visual degradation suggests that the baselines may not have been given an equally stabilized geometric starting point. A clearer description of how pose supervision, depth alignment, and initial surfaces were handled for each competing method would help ensure confidence in the reported gaps.
- All datasets are indoor and relatively structured - no explicit experiments test failure modes (irregular geometry, non-planar scenes, outdoor clutter). Standard view splits of 3, 6, and 9 views are followed in relevant literature (ReconFusion, CAT3D), but are not followed in this paper.
- A strong baseline in ViewCrafter (TPAMI'25) is missing.

**Questions:**

N/A

---

> ### Author Response · Authors · 2025-11-21
> **Response to The Reviewer (1 / 3)**
>
> We sincerely thank the reviewer for the constructive comments on our work. We greatly appreciate your positive feedback, particularly your recognition of our method’s "clear motivation" and "well-explained and technically coherent" design, as well as your acknowledgment of the results, including "fewer floaters and sharper planar regions" and "substantial gains in unobserved regions." We address each of the concerns below.
>
>
> **1. Fundamental Reliance on Planar Structures.**
>
> > Fundamental Reliance on Planar Structures: The method's core contribution and primary advantage are fundamentally tied to the Manhattan-world assumption. While effective for artificial environments, this reliance makes the approach far less suitable for organic, non-planar scenes (e.g., natural landscapes, complex statues, foliage). The paper's solution for non-planar regions is to fall back on monocular depth estimation, which is the very technique it criticizes for scale ambiguity.
>
> We would first like to note that **Manhattan-world or plane-rich environments represent a broad and practically important category of real-world scenes**, and many prior works (such as Manhattan-SDF [1], PlanarRecon [2] and IndoorGS [3]) explicitly focus on this setting due to its relevance in many important applications like indoor navigation, robotics and AR/VR. Within this widely studied and impactful domain, our method delivers substantial improvements over existing baselines, especially in under-constrained regions. We believe that achieving clear performance gains in such a common and practically meaningful scenario already constitutes a valuable contribution to the community.
>
> At the same time, we would like to clarify that our method also performs well on non-planar or less structured scenes because it is **a strict enhancement of the base model** (e.g., 2DGS or MAtCha).
> Specifically, our method handles different types of regions as follows:
> - **Planar regions (improved)**: accuracy is enhanced by leveraging plane priors for both observed and unobserved regions;
> - **Observed non-planar regions (preserved)**: the base model’s predictions are maintained to ensure reliability;
> - **Unobserved non-planar regions (improved)**: linearly aligned monocular depth is used as a fallback where the base model provides no supervision.
>
> This design ensures that G4Splat retains the general applicability of the base model while strengthening it in planar and under-constrained areas.
>
> To further demonstrate this, we have added evaluation results on all 9 scenes from Mip-NeRF 360 using 9 input views following ReconFusion [4]. We report the averaged metrics in the table below, and provide per-scene results in Appendix Table A2, along with comparisons against baselines for **all 5 outdoor scenes in Appendix Figure A5**. These results show that G4Splat **performs strongly not only on indoor, Manhattan-style scenes** (*bonsai, counter, kitchen, room*), **but also on outdoor, non-Manhattan and less structured scenes** such as *bicycle, flowers, garden, stump, and treehill*, demonstrating its robustness on non-planar environments as well. Additionally, our original submission already included compelling examples on less-planar environments, including the *Museum* in Figure 1 and *store* and *cat* in Figure 5.
>
> | Method | PSNR↑ | SSIM↑ | LPIPS↓ |
> | -------- | -------- | -------- | -------- |
> | MAtCha     | 16.44     | 0.453     | 0.389     |
> | GenFusion     | 17.82     | 0.452     | 0.439     |
> | Difix3D+     | 16.28     | 0.389     | 0.426     |
> | GuidedVD     | 16.78     | 0.438     | 0.499     |
> | See3D     | 16.92     | 0.443     | 0.451     |
> | Ours     | **18.66**     | **0.515**     | **0.371**     |
>
> Please note that the GenFusion results we report are slightly lower than those presented in its original paper. This discrepancy arises because GenFusion initializes its point cloud using the portion of the COLMAP reconstruction generated from all available views of each scene (ranging from approximately 125 to 311 views in the dataset) that are visible to the 9 input views. Such an initialization provides an almost perfectly accurate structure in the visible regions of the input viewpoints, but it also introduces additional information beyond the 9 given views. In contrast, in our experiments, we follow the same setting as in our main paper and initialize the point cloud using MASt3R-SfM with only the 9 input views.
>
> We hope these results clearly demonstrate that **G4Splat remains robust well beyond planar or Manhattan-world settings**.
>
> [1] Neural 3D Scene Reconstruction with the Manhattan-world Assumption. CVPR 2022
>
> [2] PlanarRecon: Real-time 3D Plane Detection and Reconstruction from Posed Monocular Videos. CVPR 2022
>
> [3] IndoorGS: Geometric Cues Guided Gaussian Splatting for Indoor Scene Reconstruction. CVPR 2025
>
> [4] ReconFusion: 3D Reconstruction with Diffusion Priors. CVPR 2024

---

> ### Author Response · Authors · 2025-11-21
> **Response to The Reviewer (2 / 3)**
>
> **2. Heavy engineering pipeline.**
>
> > Heavy Engineering Pipeline - G4Splat is not a single model but a complex, multi-stage pipeline that glues together numerous off-the-shelf components like MAtCha, MASt3R-SfM, SAM, K-means clustering, RANSAC, a monocular depth estimator, and a video diffusion model. This high complexity makes the system brittle - a failure in any one component could compromise the entire pipeline.
>
> Multi-stage pipelines that integrate the off-the-shelf components (e.g., MASt3R-SfM, SAM, K-means clustering, RANSAC, and monocular depth estimators) **are widely used in recent reconstruction systems**, including GuidedVD [1], Free360 [2], MAtCha [3], GenFusion [4], and NeuralPlane [5]. In this context, **G4Splat does not introduce any additional system complexity** beyond what is already standard in prior work.
>
> More importantly, G4Splat is not a simple glueing of these components. Instead, it incorporates a carefully designed geometry-guided generative training loop (Sec. 3.4) that **progressively corrects errors from earlier stages by continually integrating newly generated novel views**. This design enables the system to self-refine over time rather than being constrained by upstream inaccuracies.
>
> As a result, when using exactly **the same hyperparameters across all experiments**, G4Splat performs **consistently well** on all evaluated datasets (Replica, ScanNet++, DeepBlending, and Mip-NeRF 360) and **significantly outperforms baseline methods**. The strong and stable performance across diverse benchmarks further demonstrates the robustness of our pipeline.
>
> [1] Taming Video Diffusion Prior with Scene-Grounding Guidance for 3D Gaussian Splatting from Sparse Inputs. CVPR 2025
>
> [2] Free360: Layered Gaussian Splatting for Unbounded 360-Degree View Synthesis from Extremely Sparse and Unposed Views. CVPR 2025
>
> [3] MAtCha Gaussians: Atlas of Charts for High-Quality Geometry and Photorealism From Sparse Views. CVPR 2025
>
> [4] GenFusion: Closing the Loop between Reconstruction and Generation via Videos. CVPR 2025
>
> [5] NeuralPlane: Structured 3D Reconstruction in Planar Primitives with Neural Fields. ICLR 2025
>
>
> **3. Concern about the fairness of comparing baselines without MAtCha initialization.**
>
> > The approach inherits a strong geometric scaffold from MAtCha, including scale-aligned depth, plane priors, and reliable surface initialization. This prior stabilizes 3DGS optimization in sparse-view regimes and likely contributes to the improved geometric integrity. In contrast, several baselines (e.g., GenFusion and Difix3D+) do not assume a comparable geometric initialization and instead operate in a more challenging setting where depth must be inferred solely from generative consistency. This makes the comparison somewhat imbalanced, and it becomes difficult to isolate how much of the improvement stems from the proposed refinements versus MAtCha's initialization advantage.
>
> As stated in Sec. 4.1 of the paper and following the baseline protocol in MAtCha, **all baselines are augmented with MASt3R-SfM**, which provides a strong geometric initialization with scale-aligned depth and stable surface estimates. Therefore, all compared methods, including GenFusion and DifFix3D+, **start from a reasonably good initialization rather than inferring geometry purely from generative consistency**.
>
> To further address your concern, we additionally evaluated all 8 Replica scenes using MAtCha-initialized baselines. The results are reported below (methods marked with an asterisk and in bold denote the use of MAtCha initialization).
>
> | Method | CD↓ | F-Score↑ | NC↑ | PSNR↑ | SSIM↑ | LPIPS↓ |
> | -------- | -------- | -------- | -------- | -------- | -------- | -------- |
> | GenFusion     | 13.05     | 41.60     | 69.33     | 20.14     | 0.801     | 0.258     |
> | **GenFusion\***     | 13.63     | 36.50     | 70.18     | 19.90     | 0.797     | 0.262     |
> | Difix3D+     | 13.71     | 43.11     | 65.34     | 19.42     | 0.779     | 0.231     |
> | **Difix3D+\***     | 18.63     | 29.78     | 62.88     | 18.35     | 0.740     | 0.269     |
> | Ours     | 6.61     | 65.14     | 83.98     | 23.90     | 0.836     | 0.199     |
>
> From the table, we observe that the performance of GenFusion and DifFix3D+ even degrades slightly when switching to MAtCha initialization. This is mainly because MAtCha yields a substantially denser point cloud than the MASt3R-SfM initialization and also contains geometric inaccuracies, as illustrated in Figure 3 of the paper. **Without accurate geometric supervision**, the additional Gaussians resulting from these noisy or imprecise points **tend to become floater artifacts**, ultimately degrading reconstruction quality. In contrast, our method incorporates accurate geometry supervision, which effectively prevents such artifacts.

---

> ### Author Response · Authors · 2025-11-21
> **Response to The Reviewer (3 / 3)**
>
> The following table (using the Replica *room_0* scene as an example) compares the point-cloud sizes of MASt3R-SfM and MAtCha initializations, as well as the final number of Gaussians obtained after training with each initialization.
>
> | Method | Initialization Point Cloud Num |
> | -------- | -------- |
> | MASt3R-SfM | 601,653 |
> | MAtCha | 4,080,000 |
>
>
> | Method | Gaussian Num |
> | -------- | -------- |
> | GenFusion     | 862,637     |
> | **GenFusion\***     | 2,091,816 |
> | Difix3D+     | 943,089     |
> | **Difix3D+\***     | 4,461,395     |
>
>
> **4. How pose supervision, depth alignment, and initial surfaces were handled for each competing method?**
>
>
> > The qualitative comparisons in Fig. 4 appear to show holes and structural failures for generative baselines that are not typically reported in their original papers. This behavior is plausible when such models are applied without a metric depth prior or explicit plane constraints, especially under sparse or weakly posed conditions. However, the visual degradation suggests that the baselines may not have been given an equally stabilized geometric starting point. A clearer description of how pose supervision, depth alignment, and initial surfaces were handled for each competing method would help ensure confidence in the reported gaps.
>
> As discussed above, all baselines in our paper are initialized using MASt3R-SfM point clouds, which provide a reasonably strong geometric starting point with stable structural cues. In addition, each baseline is evaluated **using its official settings**, including its own depth supervision and its official strategy for generating novel-view camera trajectories. For all three datasets (Replica, ScanNet++, and DeepBlending), we use **the ground-truth camera poses** provided by the datasets for all baselines, so there is no risk of incorrect pose conditions affecting the results.
>
> The visual degradation observed in large missing regions is therefore not caused by weaker initialization or incorrect settings. Instead, as analyzed in Sec. 4.2 of our paper, **the performance drop is inherent to the generative baselines themselves**. Difix3D+ is primarily an image-enhancement model and is effective at reducing image artifacts but has **limited capability for completing severely missing geometry**. Meanwhile, GenFusion, See3D, and GuidedVD tend to produce **substantial Gaussian floaters in unobserved regions** due to the absence of accurate geometric supervision, which limits their ability to reconstruct large unseen areas.
>
>
> **5. Evaluation on standard view splits following relevant literature.**
>
>
> > All datasets are indoor and relatively structured - no explicit experiments test failure modes (irregular geometry, non-planar scenes, outdoor clutter). Standard view splits of 3, 6, and 9 views are followed in relevant literature (ReconFusion, CAT3D), but are not followed in this paper.
>
> In our main paper, we follow classic works such as MonoSDF [1], 3DGS, and GuidedVD, and therefore evaluate all methods on the commonly used indoor datasets Replica, ScanNet++, and DeepBlending.
>
> Additionally, following your suggestion, we conducted experiments on the Mip-NeRF 360 dataset using 9 input views, consistent with the evaluation protocol adopted in ReconFusion[2]. The corresponding results and analysis have been included above and in the revised paper.
>
>
> [1] MonoSDF: Exploring Monocular Geometric Cues for Neural Implicit Surface Reconstruction. NeurIPS 2022
>
> [2] ReconFusion: 3D Reconstruction with Diffusion Priors. CVPR 2024
>
>
> **6. A strong baseline in ViewCrafter (TPAMI'25) is missing.**
>
>
> > A strong baseline in ViewCrafter (TPAMI'25) is missing.
>
> Our baseline set already includes See3D and GuidedVD. See3D outperforms ViewCrafter as reported in the See3D paper, and GuidedVD itself is an improved variant built upon ViewCrafter. Since our method achieves consistently better results than both See3D and GuidedVD in our experiments, it is reasonable to conclude that our approach would also outperform ViewCrafter. Therefore, we did not include ViewCrafter as a separate baseline.

---

> > ### Author Response · Authors · 2025-11-28
> >
> > Thank you very much for your thoughtful comments and the time you have invested in reviewing our paper. We have provided a detailed response to the concerns you raised earlier. Could you please let us know if our clarification sufficiently addressed your questions? If anything remains unclear, we would be more than happy to continue the discussion.

---

### Author Response · Authors · 2025-11-21
**General Response**

We sincerely thank all reviewers for their time, effort, and constructive feedback. We greatly appreciate the positive recognition of our work: our paper is acknowledged as having "clear motivation, well-explained" (56rG, 3Jys), "identifies a real limitation" (HhvE), and is "reasonable and intuitive" (jjWR). All reviewers also highlighted that our results "outperform state-of-the-art baselines in both geometric accuracy and appearance quality" (56rG, HhvE, jjWR, 3Jys). We have carefully followed the comments and suggestions from all reviewers and revised our manuscript (*with changes highlighted in blue*) accordingly.

We would also like to clarify that our method performs strongly not only in planar or structured environments, but also in non-planar and less structured scenes, as it constitutes a strict enhancement of the underlying base model (e.g., 2DGS or MAtCha). To further demonstrate this, we add additional experiments on all 9 scenes from the Mip-NeRF 360 dataset using 9 input views, following the protocol of ReconFusion [1]. We report per-scene results in Appendix Table A2 and provide comparisons against baselines on all 5 outdoor scenes in Appendix Figure A5.

We hope these clarifications further strengthen your confidence in assessing our contributions. If any questions remain, we would be more than happy to address them during the discussion stage. We will continue to refine the manuscript based on all feedback, and we remain committed to delivering a high-quality paper that benefits the research community.

[1] ReconFusion: 3D Reconstruction with Diffusion Priors. CVPR 2024

---

### Meta-Review · Area_Chair_6cc8 · 2026-01-03

**Summary:**

The paper proposes G4Splat, a geometry-guided sparse-view 3D reconstruction pipeline that combines accurate planar geometry estimation with generative priors (e.g., diffusion models). The method introduces reliable metric-scale depth maps based on plane priors and integrates them into a full pipeline for visibility estimation, novel view selection, and geometry-aware inpainting. The proposed approach significantly improves both geometric fidelity and appearance quality, especially in unobserved regions, and supports real-world settings like single-view and unposed inputs.

While reviewers broadly acknowledged the significance and effectiveness of the method, several concerns were raised:

- Reviewer 56rG questioned the generality of the method, citing its reliance on planar structures and the engineering complexity of the pipeline. They also questioned the fairness of comparisons with baselines and the impact of initialization.

- Reviewer HhvE asked for better analysis of model bias, runtime, and component contributions, and raised concerns about the method’s robustness in non-planar environments.

- Reviewer jjWR suggested evaluating observed vs. unobserved regions, more generative priors, and clarifying implementation details. They also raised minor concerns about the fairness of baselines and writing clarity.

- Reviewer 3Jys raised questions about parameter choices in the visibility grid and the reliance on planar assumptions in more complex environments.

Despite these concerns, all reviewers acknowledged that the method outperforms strong baselines and provides a well-motivated integration of geometry and generative priors for high-quality 3D reconstruction.

**Reviewer Concerns:**

**Addressed Concerns**\
The rebuttal was extensive, thorough, and highly responsive. The authors provided new experiments, clarified implementation details, and addressed conceptual and empirical concerns. Key responses included:

***Robustness on non-planar scenes***: The authors added new results on Mip-NeRF 360, showing strong performance on outdoor and less-structured scenes (e.g., garden, treehill), directly addressing concerns from 56rG, HhvE, and 3Jys. These results confirm that G4Splat is not limited to Manhattan-world environments.

***Pipeline complexity***: The authors explained that their pipeline is aligned with standard practice in the field, and emphasized that G4Splat’s training loop actively refines geometry rather than relying rigidly on upstream stages. This addressed 56rG's concern about fragility and engineering overhead.

***Ablation and component contributions***: Table 2 and Section 4.3 already provided a clear ablation study. Additionally, the authors added separate evaluations for observed vs. unobserved regions, showing larger gains in unobserved regions. This addressed HhvE and jjWR.

***Generative model bias and compatibility***: The authors clarified in detail how G4Splat mitigates generative-prior bias using geometry-guided masks and view selection. They also added new experiments using alternative generative priors (ViewCrafter), showing consistent improvements, thereby addressing HhvE and jjWR.

***Fairness of baseline comparisons***: The authors clarified that all baselines were initialized with MASt3R-SfM. Additionally, they tested MAtCha + See3D, confirming that G4Splat still outperforms it. This addresses the fairness concerns (56rG, jjWR).

***Computational efficiency***: Runtime comparisons were included (Table 3), and the authors clarified the meaning of “DS” (DownSample). This addressed concerns from HhvE and 3Jys.

***Implementation details and clarity***: The paper was revised to clarify visibility grid construction, plane extrapolation, normal computation, and color supervision. These refinements addressed comments from 3Jys and jjWR.

**Remaining Minor Concerns**\
Most concerns were convincingly addressed. A few open points remain, but they are minor and do not materially detract from the paper:

- The reliance on 3D planes may still limit performance in highly organic scenes with no planar structures, although fallback mechanisms and performance on outdoor scenes mitigate this.

- The fairness of comparisons could still be debated in terms of initialization fidelity and depth supervision differences, though the authors’ empirical evaluations support their claims.

**Reviewer Scores:**

**Reviewer 56rG (Initial Score: 4)**: Despite a critical initial review, the authors addressed all major concerns with new experiments and clarifications. While the reviewer didn’t update their score, a fair reevaluation would likely raise it. Predicted Final Score: 4 or 6

**Reviewer HhvE (Initial Score: 4)**: Raised technical concerns, all of which were addressed in the rebuttal. Given the thoroughness of the response, reviewer may have raised their score. Predicted Final Score: 6

**Reviewer jjWR (Initial Score: 6)**: Provided a positive review with detailed suggestions. All were addressed. Final Score: 6 or 8

**Reviewer 3Jys (Initial Score: 6)**: Confirmed post-rebuttal that they were satisfied and kept the score. Final Score: 6

---

### Decision · Program_Chairs · 2026-01-26

Accept (Poster)